# Coefficients-Preserving Sampling for Reinforcement Learning with Flow Matching

## Abstract

Reinforcement Learning (RL) has recently emerged as a powerful technique for improving image and video generation in Diffusion and Flow Matching models, specifically for enhancing output quality and alignment with prompts. A critical step for applying online RL methods on Flow Matching is the introduction of stochasticity into the deterministic framework, commonly realized by Stochastic Differential Equation (SDE). Our investigation reveals a significant drawback to this approach: SDE-based sampling introduces pronounced noise artifacts in the generated images, which we found to be detrimental to the reward learning process. A rigorous theoretical analysis traces the origin of this noise to an excess of stochasticity injected during inference. To address this, we draw inspiration from Denoising Diffusion Implicit Models (DDIM) to reformulate the sampling process. Our proposed method, Coefficients-Preserving Sampling (CPS), eliminates these noise artifacts. This leads to more accurate reward modeling, ultimately enabling faster and more stable convergence for reinforcement learning-based optimizers like Flow-GRPO and Dance-GRPO.

## 1. Introduction

The paradigm of unsupervised pre-training, followed by supervised fine-tuning and reinforcement learning post-training, has become the new standard for training next-generation deep learning models (Achiam et al., 2023; Ouyang et al., 2022). Inspired by the application of reinforcement learning in Large Language Models (Gao et al., 2023; Rafailov et al., 2023), RL algorithms have also been adopted in the image and video generation domains (Black

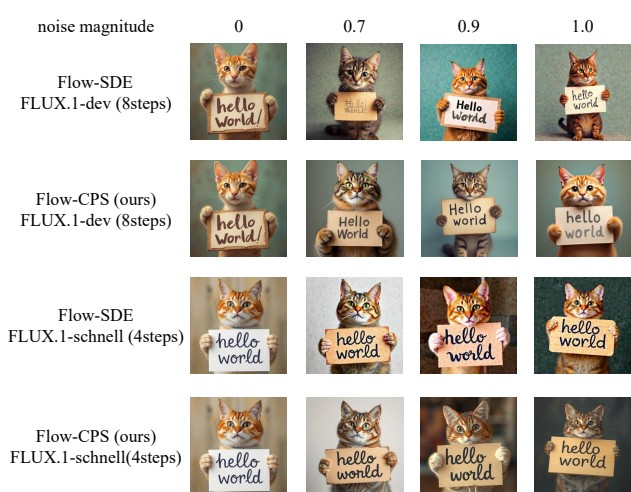

*Figure 1.* The images sampled by Flow-SDE exhibit severe noise, and the noise magnitude increases with higher sampling noise magnitude. In contrast, our Coefficients-Preserving Sampling (CPS) algorithm produces noise-free images regardless of the noise level. Notably, these images will be fed into a reward model, and the noisy images may lead to inaccurate rewards.

et al., 2023; Miao et al., 2024; Wallace et al., 2024; Dong et al., 2023; Yang et al., 2024). Recently, a series of algorithms have utilized Group Relative Policy Optimization (GRPO) to optimize for specific rewards (Liu et al., 2025; Xue et al., 2025), achieving impressive results in metrics such as aesthetics (Kirstain et al., 2023; Wu et al., 2023), instruction following (Hessel et al., 2021), and image-to-video consistency (Jiang et al., 2024).

The standard RL loop comprises three stages: sampling, reward and advantage computation, and policy optimization. A crucial requirement of the sampling stage is to generate a group of highly diverse samples for each prompt. To this end, methods such as Flow-GRPO (Liu et al., 2025) and Dance-GRPO (Xue et al., 2025) introduce stochasticity by reformulating the deterministic Ordinary Differential Equation (ODE) of the generative process as a Stochastic Differential Equation (SDE). However, we identify that during training, this SDE-based sampling produces outputs always corrupted by conspicuous noise artifacts (see Figure 1). The rewards, which guide the policy updates, are computed from these noisy samples. Consequently, reward

[1]Anonymous Institution, Anonymous City, Anonymous Region, Anonymous Country. Correspondence to: Anonymous Author <anon.email@domain.com>.

Preliminary work. Under review by the International Conference on Machine Learning (ICML). Do not distribute.

models designed to assess aesthetic quality or human preference often assign inaccurate scores and rankings, thereby misleading the learning process.

To resolve this, we thoroughly investigated the Flow-SDE sampling mechanism. Our analysis revealed that the Flow-SDE formulation injects a greater amount of noise than the original ODE. As the original ODE scheduler is retained, this excess noise accumulates, leading to a non-zero final noise level and visibly noisy outputs. Fundamentally, this problem stems from a mismatch between the SDE's score function term and the noise level introduced by the Wiener process. Inspired by DDIM (Song et al., 2021a), we reformulated the noise injection method during sampling to ensure that at every timestep, the noise level of the latent variable remains consistent with the scheduler.

We empirically validated our enhanced algorithm on multiple baseline models and with a variety of reward functions. The results confirm that for reward models predicated on aesthetics and human preferences, our method consistently demonstrates superior convergence rates and achieves higher terminal reward values. For detection-based reward models, our method achieves faster convergence rates with a similar final reward.

To summarize, our main contributions are as follows:

- We identify an issue of significant noise in images sampled via Flow-SDE (Figure 1). Through analysis, we introduce the concept of Coefficient-Preserving Sampling and prove that the original Flow-SDE fails to satisfy this requirement.

- We propose a novel sampling formulation that adheres to the Coefficient-Preserving property. By generalizing DDIM to Flow Matching, the proposed algorithm generates high-fidelity images even under high noise levels.

- We analyze the root cause of the excessive noise in Flow-SDE sampling, tracing it back to the Taylor expansion used in its derivation. We showed that this expansion not only introduces approximation errors but also induces numerical instability due to the inclusion of a $1/t$ term.

- We experimentally verify that our method significantly facilitates both reward estimation and optimization, yielding results that substantially outperform those based on Flow-SDE sampling.

## 2. Related Work

### 2.1. Alignment For Large Language Models

With the advent of Large Language Models (LLMs)(Brown et al., 2020; Achiam et al., 2023), Reinforcement Learning (RL) has garnered renewed attention. The Reinforce-

ment Learning from Human Feedback (RLHF) framework(Ouyang et al., 2022; Gao et al., 2023), for instance, trains a reward model using human preference data, which in turn fine-tunes the LLM to better align with human expectations. As an alternative to computationally intensive policy gradient methods, Direct Preference Optimization (DPO)(Rafailov et al., 2023) provides a more streamlined approach that directly trains the model on human preference data. More recently, advanced techniques have been applied to enhance multi-step reasoning. For example, OpenAI-o1 utilizes Proximal Policy Optimization (PPO)(Schulman et al., 2017) and DeepSeek-R1 employs Group Relative Policy Optimization (GRPO)(Shao et al., 2024), both using verifiable rewards to improve the models' capacity for extended reasoning via chain-of-thought.

### 2.2. Alignment for Diffusion

Analogous to autoregressive LLMs, both Diffusion(Ho et al., 2020; Song et al., 2021a;b) and Flow Matching(Lipman et al., 2022; Liu et al., 2022) models usually construct their outputs via a multi-step sampling process. They can therefore be aligned using similar RL-based techniques. As such, diffusion models are compatible with optimization algorithms including DPO(Wallace et al., 2024; Dong et al., 2023; Yang et al., 2024), PPO-style policy gradients(Black et al., 2023; Miao et al., 2024; Zhao et al., 2025), and GRPO(Liu et al., 2025; Xue et al., 2025). However, unlike token-based models that involve discrete selection steps, the absence of quantization in the diffusion process permits an alternative training paradigm: the direct backpropagation of gradients through the full sampling trajectory (Xu et al., 2023).

### 2.3. Diffusion Sampler

Efficient sampling is a critical research area for diffusion models. Samplers can be broadly categorized by the numerical methods they adapt. Foundational approaches like DDPM (Ho et al., 2020) established the paradigm but were slow. Denoising Diffusion Implicit Models (Song et al., 2021a) provided one of the first major speed improvements by formulating a deterministic sampling process. Subsequently, a significant body of work has focused on applying and adapting sophisticated ordinary differential equation (ODE) solvers. For instance, first-order methods like the Euler solver (Song et al., 2021b) offer speed at the cost of accuracy, while second-order methods like Heun's method (Karras et al., 2022) provide a better balance. High-order solvers, such as DPM-Solver (Lu et al., 2022; 2025), have become popular for their dramatic reduction in required sampling steps. Concurrent work (Zheng & Zheng, 2025) unifies all previous samplers by coefficient matrices, which are formulated by a similar rule with our proposed Coefficient-Preserving Sampling.

## 3. Preliminaries

In this section, we introduce the formulations of Flow Matching, Flow-GRPO, Dance-GRPO and DDIM. They will be the basic knowledge for our proposed algorithm.

**Flow Matching** Assume that $\boldsymbol{x}_0 \sim X_0$ is sampled from the data distribution and $\boldsymbol{x}_1 \sim X_1$ is a gaussian noise sample, Rectified Flow (Liu et al., 2022) interpolates noised sample $\boldsymbol{x}_t$ as,

$$\boldsymbol{x}_t = (1 - t)\boldsymbol{x}_0 + t\boldsymbol{x}_1, \tag{1}$$

where $t \in [0, 1]$ is the *noise level*. Then a neural network is trained to regress the velocity $\boldsymbol{v} = \boldsymbol{x}_1 - \boldsymbol{x}_0$. Finally, Flow Matching methods use a deterministic ODE for the forward process:

$$d\boldsymbol{x}_t = \hat{\boldsymbol{v}}_\theta(\boldsymbol{x}_t, t)dt, \tag{2}$$

where $\hat{\boldsymbol{v}}_\theta(\boldsymbol{x}_t, t)$ is the estimated velocity. The *hat* $(\hat{\cdot})$ denotes that the value is model predicted in the following article.

**Flow-GRPO and Dance-GRPO** Reinforcement Learning relies on stochastic sampling to generate diverse samples. Flow-GRPO (Liu et al., 2025) and Dance-GRPO (Xue et al., 2025) introduce randomness into Flow Matching by converting the deterministic Flow-ODE into Flow-SDE,

$$d\boldsymbol{x}_t = [\boldsymbol{v}_\theta(\boldsymbol{x}_t, t) + \frac{\sigma_t^2}{2t}(\boldsymbol{x}_t + (1-t)\hat{\boldsymbol{v}}_\theta(\boldsymbol{x}_t, t))]dt + \sigma_t\sqrt{dt}\boldsymbol{\epsilon}, \tag{3}$$

where $\boldsymbol{\epsilon} \sim \mathcal{N}(0, \boldsymbol{I})$ is a newly sampled gaussian noise, $\sigma_t = \eta\sqrt{\frac{t}{1-t}}$ for Flow-GRPO and $\sigma_t = \eta$ for Dance-GRPO.

After sampling a group of $G$ diverse images $\{\boldsymbol{x}_0^i\}_{i=1}^G$, the rewards $R(\boldsymbol{x}_0^i)$ are transformed to advantages by,

$$A_t^i = \frac{R(\boldsymbol{x}_0^i) - \text{mean}(\{R(\boldsymbol{x}_0^i)\}_{i=1}^G)}{\text{std}(\{R(\boldsymbol{x}_0^i)\}_{i=1}^G)}. \tag{4}$$

Then GRPO (Shao et al., 2024) optimizes the policy model by maximizing the following objective,

$$\mathcal{L}(\theta) = \mathbb{E}_{\boldsymbol{x}^i \sim \pi_{\theta_{\text{old}}}} \frac{1}{G} \sum_{i=1}^G \frac{1}{T} \sum_{t=0}^{T-1} \left( min \left( r_t^i(\theta)A_t^i, \right.\right.$$

$$\left.\left. \text{clip}(r_t^i(\theta), 1-\epsilon, 1+\epsilon)A_t^i \right) - \beta D_{\text{KL}}(\pi_\theta || \pi_{\text{ref}}) \right), \tag{5}$$

where $r_t^i(\theta) = \frac{p_\theta(\boldsymbol{x}_{t-\Delta t}^i | \boldsymbol{x}_t^i)}{p_{\theta_{\text{old}}}(\boldsymbol{x}_{t-\Delta t}^i | \boldsymbol{x}_t^i)}$ and the KL loss term is defined as a closed form:

$$D_{\text{KL}}(\pi_\theta || \pi_{\text{ref}}) = \frac{\|\overline{\boldsymbol{x}}_{t-\Delta t, \theta} - \overline{\boldsymbol{x}}_{t-\Delta t, \text{ref}}\|^2}{2\sigma_t^2\Delta t}, \tag{6}$$

where $\overline{\boldsymbol{x}}$ denotes the mean of predicted $\boldsymbol{x}$, which is implemented by removing the injected noise.

**DDPM and DDIM Sampling** SDE is not the only way to inject stochasticity. In the DDIM sampling procedure,

$$\boldsymbol{x}_{t-1} = \sqrt{\alpha_{t-1}} \underbrace{\left( \frac{\boldsymbol{x}_t - \sqrt{1-\alpha_t}\epsilon_\theta^{(t)}(\boldsymbol{x}_t)}{\sqrt{\alpha_t}} \right)}_{\text{predicted } \boldsymbol{x}_0} +$$

$$\sqrt{1 - \alpha_{t-1} - \sigma_t^2} \cdot \underbrace{\epsilon_\theta^{(t)}(\boldsymbol{x}_t)}_{\text{predicted noise}} + \underbrace{\sigma_t\epsilon_t}_{\text{random noise}}, \tag{7}$$

where $\epsilon_t \sim \mathcal{N}(0, \boldsymbol{I})$ is standard Gaussian noise independent of $\epsilon_\theta^{(t)}(\boldsymbol{x}_t)$. For other notations, please refer to the DDIM paper (Song et al., 2021a). When $\sigma_t = \sqrt{(1-\alpha_{t-1})/(1-\alpha_t)}\sqrt{1-\alpha_t/\alpha_{t-1}}$, the forward process becomes Markovian, and the generative process becomes a DDPM. When $\sigma_t = 0$, the resulting model becomes an implicit probabilistic model (DDIM). For other $\sigma_t$, we call it DDIM with stochasticity.

The relationship between DDIM and DDPM is similar to that between ODE and SDE: DDIM and ODE are deterministic, while DDPM and SDE inject stochasticity into their counterparts.

## 4. Analysis and Methods

In this section, we first introduce the concept of coefficients-preserving sampling (CPS). Then we prove that the SDE used in Flow-GRPO and Dance-GRPO cannot match the requirements of CPS. Finally, we provide an alternative to SDE to inject stochasticity for flow matching.

### 4.1. Coefficients-Preserving Sampling

During the sampling process of flow matching, we can get the predicted sample $\hat{\boldsymbol{x}}_0$ and noise $\hat{\boldsymbol{x}}_1$ by,

$$\hat{\boldsymbol{x}}_0 = \boldsymbol{x}_t - t\hat{\boldsymbol{v}}, \ \ \hat{\boldsymbol{x}}_1 = \boldsymbol{x}_t + (1-t)\hat{\boldsymbol{v}}. \tag{8}$$

Referring to Equation 8, we can rewrite the Flow-ODE sampling function as,

$$\hat{\boldsymbol{x}}_{t-\Delta t} = \boldsymbol{x}_t - \hat{\boldsymbol{v}}_\theta(\boldsymbol{x}_t, t)\Delta t$$

$$= (1 - (t - \Delta t)) \underbrace{(\boldsymbol{x}_t - t\hat{\boldsymbol{v}}_\theta(\boldsymbol{x}_t, t))}_{\text{predicted } \hat{\boldsymbol{x}}_0} +$$

$$(t - \Delta t) \underbrace{(\boldsymbol{x}_t + (1-t)\hat{\boldsymbol{v}}_\theta(\boldsymbol{x}_t, t))}_{\text{predicted } \hat{\boldsymbol{x}}_1}$$

$$= \underbrace{(1 - (t - \Delta t))}_{\text{coefficient of sample}} \hat{\boldsymbol{x}}_0 + \underbrace{(t - \Delta t)}_{\text{coefficient of noise}} \hat{\boldsymbol{x}}_1, \tag{9}$$

which is also a linear interpolation between the *predicted* sample $\hat{\boldsymbol{x}}_0$ and *predicted* noise $\hat{\boldsymbol{x}}_1$. This equation reveals that the sum of the coefficients for the sample and the noise is always 1, whether for training or inference. If this condition is not satisfied, the out-of-distribution input to the

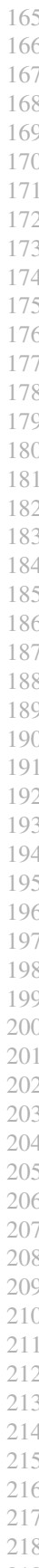
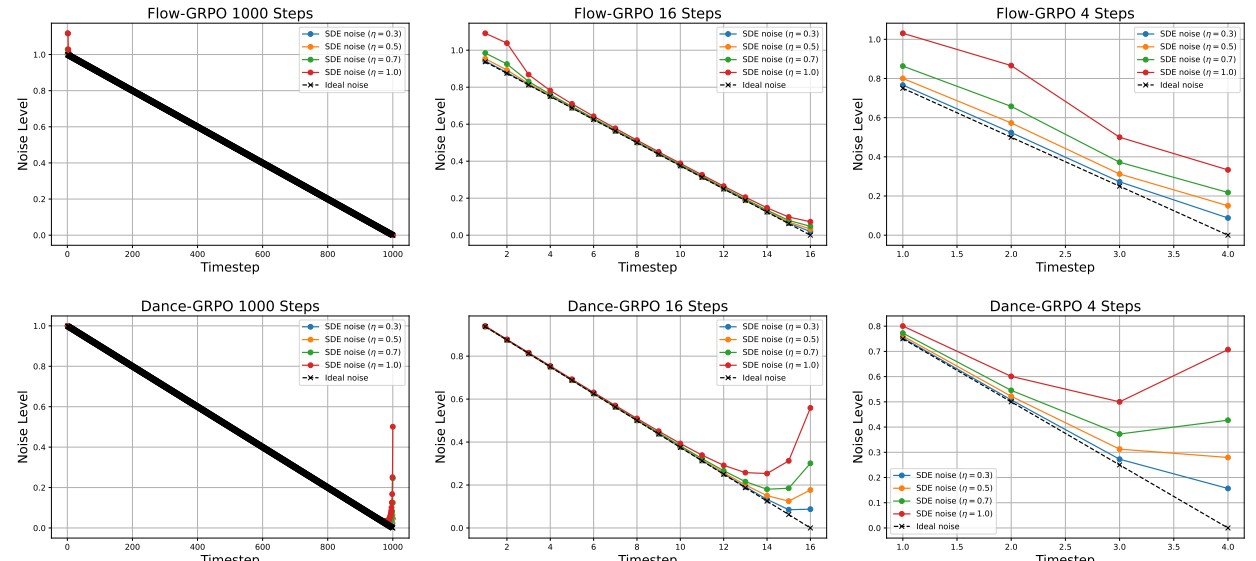

*Figure 2.* The ideal noise level $t$ and SDE noise level (Equation 11) for Flow-GRPO and Dance-GRPO with 1000, 16, and 4 sampling steps. Except for the numerical problem around $t = 0$ and $t = 1$, the error of the noise level increases as the sampling step decreases.

neural network will potentially yield an incorrect velocity field.

Furthermore, the sampling process usually utilizes a scheduler that strictly defines the target $t$ for each step. Denoising too much or too little at any timestep will distort the final generated image. Based on the preceding analysis, we define coefficients-preserving sampling as follows:

**Definition 4.1** (Coefficients-Preserving Sampling). A sampling process is considered to be **coefficients preserving** if it satisfies the following two conditions:

1. The coefficient of the sample should be strictly allocated by the scheduler for all timesteps.
2. The total noise level, defined as the standard deviation of a single multivariate noise or the root sum square (RSS) of the standard deviations of multiple independent noises, must align with the scheduler for all timesteps.

The definition of the total noise level relies on two key assumptions. First, we assume that the predicted noise terms, denoted as $\epsilon_\theta$ in Equation 7 or $\hat{x}_1$ in Equation 8, adhere to the properties of a standard Gaussian distribution, e.g., zero mean and unit variance. This is a standard assumption in diffusion sampling algorithms (Song et al., 2021a; Karras et al., 2022; Lu et al., 2022), given that they necessitate replacing ground truth variables with predicted variables during inference, despite the gap between training and testing. Second, we assume that the predicted noise is statistically independent of the newly injected noise $\epsilon_t$. This independence holds by construction, since $\epsilon_t$ is explicitly sampled from a fresh,

independent Gaussian distribution at each timestep.

**DDIM sampling is Coefficients-Preserving:** In Equation 7, there are two independent noise terms, whose coefficients are $\sqrt{1 - \alpha_{t-1} - \sigma_t^2}$ and $\sigma_t$, so the final noise level is their RSS $\sqrt{1 - \alpha_{t-1}}$. The sample coefficient is $\sqrt{\alpha_{t-1}}$, so the squared sum of the two coefficients is 1. These two coefficients exactly match the DDIM scheduler, whatever $\sigma_t$ is. Thus, we say the sampling procedure of DDIM is Coefficients-Preserving Sampling.

### 4.2. Flow-SDE is not Coefficients-Preserving Sampling

Recall Equation 3 and rewrite it into the similar form of Equation 7,

$$\boldsymbol{x}_{t-\Delta t}$$
$$= \boldsymbol{x}_t - [\hat{\boldsymbol{v}}_\theta(\boldsymbol{x}_t, t) + \frac{\sigma_t^2}{2t}\underbrace{(\boldsymbol{x}_t + (1-t)\hat{\boldsymbol{v}}_\theta(\boldsymbol{x}_t, t))}_{\text{predicted } \hat{\boldsymbol{x}}_1}]\Delta t + \sigma_t\sqrt{\Delta t}\boldsymbol{\epsilon}$$
$$= \underbrace{\boldsymbol{x}_t - \hat{\boldsymbol{v}}_\theta(\boldsymbol{x}_t, t)\Delta t}_{\text{Equation 9}} - \frac{\sigma_t^2 \Delta t}{2t}\hat{\boldsymbol{x}}_1 + \sigma_t\sqrt{\Delta t}\boldsymbol{\epsilon}$$
$$= (1 - (t - \Delta t))\,\hat{\boldsymbol{x}}_0 + (t - \Delta t - \frac{\sigma_t^2 \Delta t}{2t})\hat{\boldsymbol{x}}_1 + \sigma_t\sqrt{\Delta t}\boldsymbol{\epsilon}. \quad (10)$$

From the above equation, we can infer that the total noise level,

$$\sigma_{total} = \sqrt{(t - \Delta t - \frac{\sigma_t^2 \Delta t}{2t})^2 + \sigma_t^2 \Delta t}$$
$$= \sqrt{(t - \Delta t)^2 + \frac{(\sigma_t \Delta t)^2}{t} + (\frac{\sigma_t^2 \Delta t}{2t})^2}$$
$$\geq t - \Delta t, \quad (11)$$

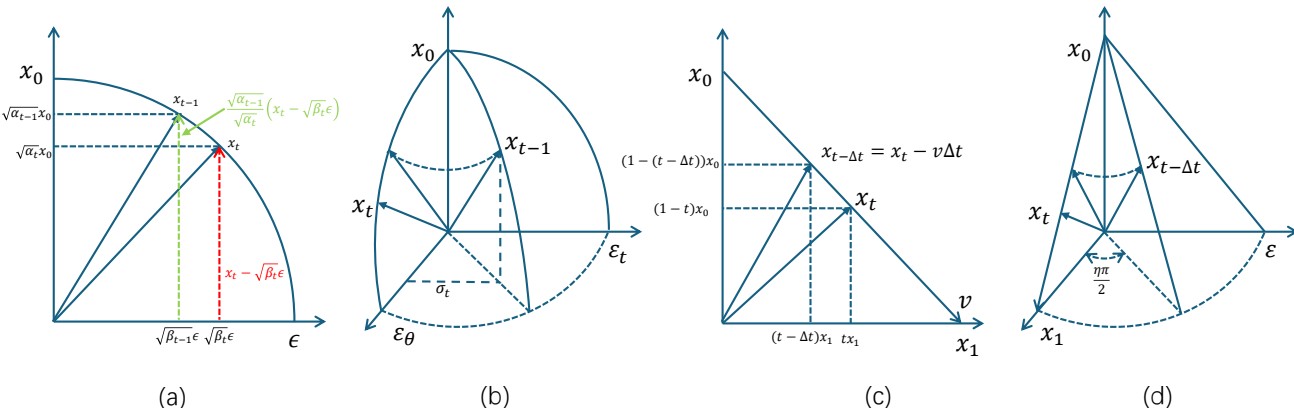

Figure 3. **(a)**: DDIM deterministic sampling process. Note that $\epsilon$ is a random Gaussian noise, which is almost orthogonal to the sample $\boldsymbol{x}_0$. Since $\sqrt{\alpha_t}^2 + \sqrt{\beta_t}^2 = 1$, the trajectory is part of a quarter-circle at each step. **(b)**: DDIM sampling process with stochasticity (Equation 7). $\epsilon_t$ is also a random Gaussian noise, which is almost orthogonal to $\boldsymbol{x}_0$ and $\epsilon_\theta$. **(c)**: Flow matching Euler Sampler. The trajectory is a straight line at each step. **(d)**: Our proposed Coefficients-Preserving Sampling (Equation 13).

where the equality holds only if $\sigma_t = 0$, which means no stochasticity. Thus, Flow-SDE cannot satisfy the second condition of CPS. At each timestep $t$, it mixes a higher level of noise into the latent variable $\boldsymbol{x}_{t-\Delta t}$, which would cause a wrong velocity direction, and the final sampled image would be noisy as shown in Figure 1.

In Figure 2, we plot the total noise level for both Flow-GRPO and Dance-GRPO. As we can see, the noise level mismatch problem is severe for both of them. Moreover, because of the $\frac{\sigma_t^2 \Delta t}{2t}$ term in Equation 10, the error around $t = 0$ is large for Dance-GRPO ($\sigma_t = \eta$). For Flow-GRPO, $\sigma_t = \eta\sqrt{\frac{t}{1-t}}$, the noise level is inaccurate around $t = 1$. The problem becomes even worse when the sampling step is low, e.g. 4 steps for FLUX.1-schnell.

### 4.3. Our Solution

The main problem of Flow-SDE is that the reduced noise level $\frac{\sigma_t^2 \Delta t}{2t}$ cannot match the newly added noise level $\sigma_t\sqrt{\Delta t}$. Noticing that DDIM also injects noise into the sampling procedure while preserving the noise level (Figure 3.b), we consider referring to DDIM sampling to solve the problem. Assume that the newly added noise has a variance of $\sigma_t^2$, the coefficient of predicted noise should be $\sqrt{(t - \Delta t)^2 - \sigma_t^2}$ to meet the requirement of the second condition of CPS. In this way, the sampling formulation is,

$$\boldsymbol{x}_{t-\Delta t} = (1 - (t - \Delta t))\,\hat{\boldsymbol{x}}_0 + \sqrt{(t - \Delta t)^2 - \sigma_t^2}\hat{\boldsymbol{x}}_1 + \sigma_t\boldsymbol{\epsilon},$$
$$(12)$$

which has a very similar form to DDIM with stochasticity (Equation 7).

For the injected noise level $\sigma_t$, the maximum value is $t - \Delta t$, or the *sqrt* term would have a negative radicand. To avoid the negative radicand, we propose to set $\sigma_t = (t -$

$\Delta t)\sin(\frac{\eta\pi}{2})$. Then the sampling formulation becomes,

$$\boldsymbol{x}_{t-\Delta t} = (1 - (t - \Delta t))\,\hat{\boldsymbol{x}}_0 +$$
$$(t - \Delta t)\cos(\frac{\eta\pi}{2})\hat{\boldsymbol{x}}_1 + (t - \Delta t)\sin(\frac{\eta\pi}{2})\boldsymbol{\epsilon}, \quad (13)$$

where $\eta \in [0, 1]$ controls the stochastic strength. This formulation satisfies the requirement of CPS and has an intuitive geometric interpretation as shown in Figure 3.d. Because our sampling algorithm is based on the CPS, we name it as Flow-CPS.

To train with GRPO, we also need $p_\theta(\boldsymbol{x}_{t-1}^i|\boldsymbol{x}_t^i)$, which is defined as (Liu et al., 2025),

$$\log p_\theta(\boldsymbol{x}_{t-\Delta t}^{old}|\boldsymbol{x}_t^{old}) = -\frac{\|\boldsymbol{x}_{t-\Delta t}^{old} - \mu_\theta(\boldsymbol{x}_t^{old}, t)\|^2}{2\sigma_t^2}$$
$$- \log\sigma_t - \log\sqrt{2\pi}, \quad (14)$$

where $\mu_\theta(\boldsymbol{x}_t, t) = (1 - (t - \Delta t))\,\hat{\boldsymbol{x}}_0 + (t - \Delta t)\cos(\frac{\eta\pi}{2})\hat{\boldsymbol{x}}_1$ in our case. For each step, the $-\log\sigma_t - \log\sqrt{2\pi}$ is a constant value that cancels out in $r_t^i(\theta) = \frac{p_\theta(\boldsymbol{x}_{t-\Delta t}^i|\boldsymbol{x}_t^i)}{p_{\theta_{old}}(\boldsymbol{x}_{t-\Delta t}^i|\boldsymbol{x}_t^i)}$. Moreover, we removed the $\sigma_t$ in the denominator to avoid division by zero or very small values in the last timestep. Thus, our definition of log-probability is as simple as,

$$\log p_\theta(\boldsymbol{x}_{t-\Delta t}^{old}|\boldsymbol{x}_t^{old}) = -\|\boldsymbol{x}_{t-\Delta t}^{old} - \mu_\theta(\boldsymbol{x}_t^{old}, t)\|^2. \quad (15)$$

Analytically, the normalization term $2\sigma_t^2$ disproportionately emphasizes the optimization of later timesteps, which involve less stochasticity. Removing this term reallocates greater weight to earlier timesteps, which typically exhibit higher diversity, so the convergence speed will accelerate.

Meanwhile, the denominator in the KL loss function (Equation 6) should also be removed:

$$D_{\mathrm{KL}}(\pi_\theta||\pi_{\mathrm{ref}}) = \|\mu_\theta(\boldsymbol{x}_t) - \mu_{ref}(\boldsymbol{x}_t)\|^2. \quad (16)$$

## 4.4. Discussion

In Equation 12, we choose not to build a Wiener process as our objective is to incorporate sufficient stochasticity to produce more diverse samples. To create a Wiener process, just replace $\sigma_t$ with $\sigma_t\sqrt{\Delta t}$,

$$\boldsymbol{x}_{t-\Delta t} = (1 - (t - \Delta t))\,\hat{\boldsymbol{x}}_0 + \\ \sqrt{(t-\Delta t)^2 - \sigma_t^2 \Delta t}\,\hat{\boldsymbol{x}}_1 + \sigma_t\sqrt{\Delta t}\boldsymbol{\epsilon}. \quad (17)$$

We name this sampling function Flow-CPWS, where W denotes the Wiener process.

Inspired by the derivation of VP-SDE ( (Song et al., 2021b)), which uses Taylor expansion for formula derivation, we can also get an approximate SDE from Flow-CPWS. Note that by Taylor expansion, $\sqrt{t^2 - x} = t - \frac{x}{2t} + O(x^2)$ around $x = 0$, the above equation can be transformed to,

$$\boldsymbol{x}_{t-\Delta t}$$
$$\approx (1-(t-\Delta t))\,\hat{\boldsymbol{x}}_0 + \left(t - \Delta t - \frac{\sigma_t^2 \Delta t}{2(t-\Delta t)}\right)\hat{\boldsymbol{x}}_1 + \sigma_t\sqrt{\Delta t}\boldsymbol{\epsilon}$$
$$\approx (1-(t-\Delta t))\,\hat{\boldsymbol{x}}_0 + \left(t - \Delta t - \frac{\sigma_t^2 \Delta t}{2t}\right)\hat{\boldsymbol{x}}_1 + \sigma_t\sqrt{\Delta t}\boldsymbol{\epsilon},$$
$$(18)$$

which is the same with Flow-SDE (Equation 10). The approximate equality holds when $\sigma_t\sqrt{\Delta t} \ll t - \Delta t$ and $\Delta t \to 0$[1]. Now we can conclude:

**Theorem 4.2.** *Flow-SDE is a first-order Taylor approximation of Flow-CPWS in the limit of $\sigma_t\sqrt{\Delta t} \ll t - \Delta t$ and $\Delta t \to 0$, with a noise level error of $\sqrt{\frac{(\sigma_t \Delta t)^2}{t} + (\frac{\sigma_t^2 \Delta t}{2t})^2}$.*

The proof is provided above by Equation 18 and Equation 11.

For traditional diffusion methods, such as DDPM (Ho et al., 2020), the sampling step is set as 1000, so the condition $\Delta t \to 0$ is well satisfied. However, for modern diffusion and flow matching samplers (Song et al., 2021a; Lu et al., 2022), the sampling step is usually less than 20. With some distillation techniques (Song et al., 2023; Yin et al., 2024), the sampling step can be reduced to 4 or even 1. The condition $\Delta t \to 0$ no longer holds in these settings. This is the fundamental reason why Flow-SDE produces inaccurate noise levels.

Furthermore, because of the $\frac{1}{t}$ term introduced by the Taylor expansion, the approximation error will be significant around $t = 0$ (Figure 2). Consequently, the approximation remains inaccurate even with a high number of sampling steps. We provide an alternative to mitigate this issue in Appendix C.

---

[1] $\Delta t \to 0$ does not necessarily mean $\sigma_t\sqrt{\Delta t} \ll t - \Delta t$, since $t - \Delta t$ can be very small in the last few steps. $\sigma_t$ must also be bounded relative to $t - \Delta t$.

## 5. Experiments

In this section, we will evaluate the performance of Flow-CPS in the circumstance of GRPO-based reward optimization on four reward models, GenEval (Ghosh et al., 2023), Text Rendering OCR (Cui et al., 2025), PickScore (Kirstain et al., 2023) and HPSv2 (Wu et al., 2023).

### 5.1. Experimental Setup

To make the experiments more convincing, we do experiments on two baselines, Flow-GRPO (Liu et al., 2025) and Dance-GRPO (Xue et al., 2025). We follow their experimental settings and only change the sampling method and the log-probability. All the experiments are conducted on $8\times$ NVIDIA A100 GPUs. We introduce two kinds of tasks, verifiable rewards (RLVR) and preference rewards (RLHF), to evaluate our proposed method.

**RLVR** Following Flow-GRPO, we use two kinds of verifiable rewards, GenEval and OCR. The GenEval is an object-focused framework to evaluate compositional image properties such as object co-occurrence, position, count, and color. The GenEval rewards are rule-based: (1) **Counting:** $r = 1 - |N_{gen} - N_{ref}|/N_{ref}$; (2) **Position/Color:** If the object count is correct, a partial reward is assigned; the remainder is granted when the predicted position or color is also correct.

The OCR reward relies on an OCR model to recognize text from the generated images and compare them with given prompts. The reward value is $r = \max(1 - N_e/N_{ref}, 0)$, where $N_e$ is the minimum edit distance between the rendered text and target text and $N_{ref}$ is the number of characters inside the quotation marks in the prompt.

**RLHF** An alternative paradigm for reward modeling is rooted in human preferences, exemplified by models like PickScore and HPSv2. The process begins with humans scoring a set of sampled images to create a preference dataset. Following this, a regression head is trained atop a foundation model, commonly the CLIP encoder, to fit these human scores. Once trained, this model serves as a direct scoring function to assess image quality.

The KL loss weight $\beta$ is a key hyperparameter in Diffusion-RL to alleviate reward hacking; we exclude it from most experiments due to its negative impact on training speed. The exception is the GenEval task, where we experimentally find that omitting the KL loss degraded performance. After careful tuning, we ultimately set $\beta = 0.001$ for our algorithm on GenEval.

### 5.2. Clean Image Sampling

As illustrated in Figure 1, our Flow-CPS consistently generates diverse and noise-free images, even at high noise levels. Conversely, the images generated by Flow-SDE

*Table 1.* GenEval Results on base model (Esser et al., 2024) and base code (Liu et al., 2025)

| Model | Overall | Single Obj. | Two Obj. | Counting | Colors | Position | Attr. Binding |
|---|---|---|---|---|---|---|---|
| SD3.5-M (base model) | 0.63 | 0.98 | 0.78 | 0.50 | 0.81 | 0.24 | 0.52 |
| +Flow-GRPO wo/ KL | 0.95 | 0.99 | 0.98 | 0.95 | 0.92 | 0.95 | 0.83 |
| +Flow-CPS wo/ KL | 0.94 | 0.99 | 0.95 | 0.95 | 0.89 | 0.93 | 0.83 |
| +Flow-GRPO w/ KL | 0.97 | 1.00 | 1.00 | 0.97 | 0.94 | 0.98 | 0.90 |
| +Flow-CPS w/ KL | 0.97 | 1.00 | 0.99 | 0.95 | 0.94 | 0.98 | 0.93 |

*Table 2.* PickScore Results

| Model | PickScore |
|---|---|
| FLUX.1-schnell | 21.86 |
| +Flow-GRPO | 23.39 |
| +Flow-CPS(ours) | 23.78 |
| FLUX.1-dev | 22.06 |
| +Flow-GRPO | 23.90 |
| +Flow-CPS(ours) | 24.25 |

*Table 3.* HPSv2 Results

| Model | HPSv2 |
|---|---|
| FLUX.1-schnell | 0.304 |
| +Dance-GRPO | 0.364 |
| +Flow-CPS(ours) | 0.377 |

*Table 4.* OCR Results. 5 repeated trials calculate errors.

| Model | OCR |
|---|---|
| SD3.5-M | 0.579 |
| +Flow-GRPO | $0.967 \pm 0.003$ |
| +Flow-CPS(ours) | $0.972 \pm 0.004$ |

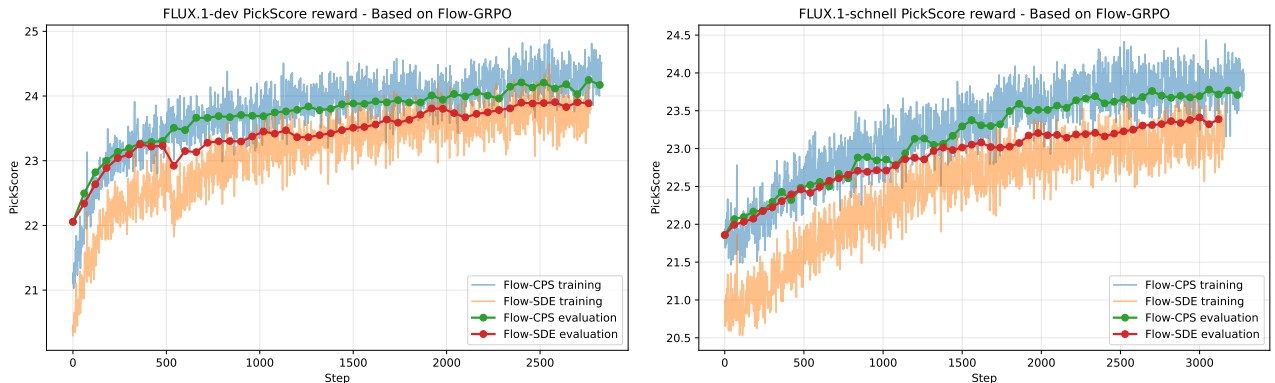

*Figure 4.* **Left**: PickScore optimization based on FLUX.1-dev. The sampling step number is 6 for training and 28 for evaluation. **Right**: PickScore optimization based on FLUX.1-schnell. The sampling step number is 4 for both training and evaluation. Note that there is no stochasticity during evaluation, so the rewards of the two sampling methods are the same at the beginning. For all experiments, we set $\eta = 0.9$.

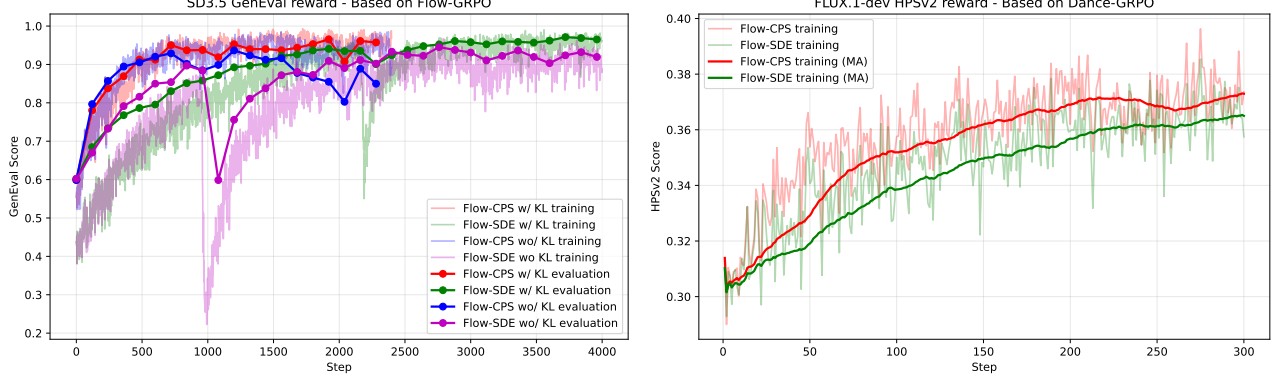

*Figure 5.* **Left**: GenEval optimization based on SD3.5. The sampling step number is 10 for training and 40 for evaluation. It is crucial to note that the exclusion of the KL loss resulted in significant performance degradation or model collapse for both sampling methods. We set $\eta = 0.7$ in these experiments. **Right**: HPSv2 optimization based on FLUX.1-dev. Since the codebase of Dance-GRPO does not provide online evaluation, we show the moving average of the training curves and leave the final evaluation performance in Table 3. We set $\eta = 0.7$ for our method and $\eta = 0.3$ (default value) for Dance-GRPO.

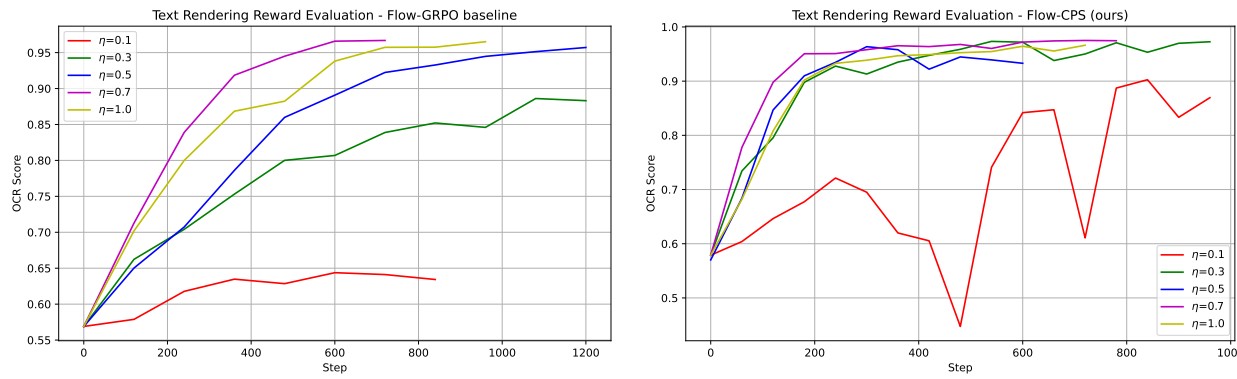

*Figure 6.* Text Rendering Reward comparison. Note that Flow-CPS (ours) converges faster than the Flow-GRPO baseline.

suffer from obvious noise, particularly under high noise conditions, which contributes to less reliable reward calculations. This observation is corroborated by Figure 4, which shows that Flow-CPS achieves higher rewards than Flow-SDE early in the training process. Furthermore, since the generation process is deterministic (no noise) at inference time, Flow-SDE also suffers from a more significant train-test discrepancy than Flow-CPS.

### 5.3. RL Results

We present the results of our method on the GenEval, PickScore, HPSv2, and OCR tasks in Table 1, 2, 3, 4, respectively. On PickScore, HPSv2 and OCR, our method consistently outperforms the two baseline methods, Flow-GRPO and Dance-GRPO. For the GenEval task, we achieve a result on par with the baselines, as the performance is already nearing saturation. However, as shown in Figure 5, our method converges to the optimal result at a faster speed, demonstrating our algorithm's advantage.

The baselines reported in Table 1, 2, 3, 4 employ the log-prob definition (Equation 14) from their respective original papers. In contrast, for Flow-CPS, we adopt the formulation in Equation 15.

For the OCR task, we conduct 5 repeated experiments to calculate errors for the baseline and our proposed method.

### 5.4. Ablation Study

**Hyperparameter $\eta$.** We conducted an ablation study on the hyperparameter $\eta$ in Equation 13, presenting the results in Figure 6. From the figure, we can conclude that our method converges significantly faster than the Flow-GRPO baseline. Both our method and the baseline method achieve their best performance when $\eta = 0.7$, while neither method can converge properly when $\eta = 0.1$.

**Denominator in LogProb.** In Equation 15, we removed the denominator $2\sigma_t^2$ to prevent numerical instability caused by division by near-zero values in the final diffusion steps. We

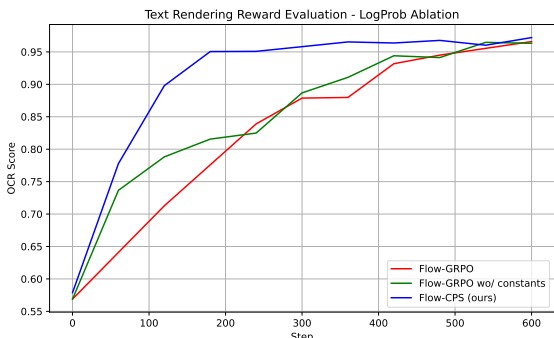

*Figure 7.* The ablation on the log-probability. Our algorithm fails to converge with the denominator $2\sigma_t^2$, so it is not shown in this figure. Removing the denominator from the baseline initially accelerates reward optimization. However, due to the noisy image issue, the reward growth curve quickly aligns with that of the baseline.

also applied this modification to the Flow-GRPO baseline for an ablation study. As shown in Figure 7, although this change initially accelerates convergence, the final performance is comparable to the original version because of the excess noise in the sampled images.

### 6. Conclusion

This paper introduces Coefficients-Preserving Sampling (CPS), a method that successfully addresses the image noise problem inherent in SDE-based sampling. Our theoretical analysis reveals that SDE is a first-order Taylor approximation of CPS. Even under conditions of extremely high noise, CPS is capable of generating diverse and clean image samples. Consequently, reward optimization guided by CPS surpasses SDE-based approaches on a variety of tasks.

Nevertheless, current Flow Matching-based GRPO methods still suffer from several unresolved issues that warrant further research. Key challenges include vulnerability to reward hacking, the credit assignment problem in multi-step exploration, and an inability to optimize for the stochasticity arising from input noise.

## Impact Statement

This paper presents work whose goal is to advance the field of Machine Learning. There are many potential societal consequences of our work, none which we feel must be specifically highlighted here.

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

## A. VP-SDE is an Approximation of DDPM

Similar to Theorem 4.2, VP-SDE (Song et al., 2021b) can also be seen as a first-order Taylor approximation of DDPM. In the VP-SDE, the forward process is,

$$
\begin{aligned}
\boldsymbol{x}(t + \Delta t) &= \sqrt{1 - \beta(t + \Delta t)\Delta t}\; \boldsymbol{x}(t) + \sqrt{\beta(t + \Delta t)\Delta t}\; \boldsymbol{z}(t) \\
&\approx \boldsymbol{x}(t) - \frac{1}{2}\beta(t + \Delta t)\Delta t\, \boldsymbol{x}(t) + \sqrt{\beta(t + \Delta t)\Delta t}\; \boldsymbol{z}(t) \\
&\approx \boldsymbol{x}(t) - \frac{1}{2}\beta(t)\Delta t\, \boldsymbol{x}(t) + \sqrt{\beta(t)\Delta t}\; \boldsymbol{z}(t),
\end{aligned}
\tag{19}
$$

where the approximate equality holds when $\Delta t \ll 1$. Similar to formula 18, it uses Taylor expansion and omits the second and higher order terms. It is interesting to note that Equation 19 is very similar to Equation 18, which also uses Taylor expansion in the first approximation and omits a $\Delta t$ in the second approximation.

For the VP-SDE backward process, please refer to Appendix E of (Song et al., 2021b), which also uses Taylor expansion and omits high-order terms in the derivation. (Song et al., 2021b) claimed that the ancestral sampler of DDPM is essentially a discretization of the reverse-time SDE. Conversely, if a pre-trained DDPM is given, we can also say that the reverse process of the VP-SDE is a continuous approximation of DDPM ancestral sampling.

## B. DPM-Solver series

DPM-Solver (Lu et al., 2022) and its variants DPM-Solver++ series(Lu et al., 2025), also provide SDE solvers. In this section, we will verify if they meet the requirements of CPS.

For SDE-DPM-Solver-1, the sampling function is,

$$
\boldsymbol{x}_t = \frac{\alpha_t}{\alpha_s}\boldsymbol{x}_s - 2\sigma_t(e^h - 1)\hat{\boldsymbol{x}}_1 + \sigma_t\sqrt{e^{2h} - 1}\boldsymbol{\epsilon},
\tag{20}
$$

where $e^h = \frac{\alpha_t}{\sigma_t}\frac{\sigma_s}{\alpha_s}$, $\alpha_t = 1 - t$ and $\sigma_t = t$ in the concept of Flow Matching. Reformulate it into the factorized form,

$$
\boldsymbol{x}_t = \alpha_t\hat{\boldsymbol{x}}_0 + \left(\frac{\alpha_t}{\alpha_s}\sigma_s - 2\sigma_t(e^h - 1)\right)\hat{\boldsymbol{x}}_1 + \sigma_t\sqrt{e^{2h} - 1}\boldsymbol{\epsilon}.
\tag{21}
$$

The coefficient of sample is $\alpha_t$, which exactly matches the first condition of CPS. However, the total noise level is,

$$
\begin{aligned}
\sigma_{total} &= \sqrt{\left(\frac{\alpha_t}{\alpha_s}\sigma_s - 2\sigma_t(e^h - 1)\right)^2 + \sigma_t^2(e^{2h} - 1)} \\
&= \sqrt{(\sigma_t e^h - 2\sigma_t(e^h - 1))^2 + \sigma_t^2(e^{2h} - 1)} \\
&= \sigma_t\sqrt{(2 - e^h)^2 + (e^{2h} - 1)} \\
&= \sigma_t\sqrt{2e^{2h} - 4e^h + 3} \\
&= \sigma_t\sqrt{2(e^h - 1)^2 + 1} \\
&\geq \sigma_t,
\end{aligned}
\tag{22}
$$

where the equality holds only when $e^h = 1$, so the SDE-DPM-Solver-1 is not Coefficient-Preserving Sampling.

For SDE-DPM-Solver++1, the sampling function is,

$$
\boldsymbol{x}_t = \frac{\sigma_t}{\sigma_s}e^{-h}\boldsymbol{x}_s + \alpha_t(1 - e^{-2h})\hat{\boldsymbol{x}}_0 + \sigma_t\sqrt{1 - e^{-2h}}\boldsymbol{\epsilon}.
\tag{23}
$$

*Table 5.* OCR Results for SDE-DPM-Solver++1 and Flow-CPS

| Model | DPM++ run1 | DPM++ run2 | CPS $\eta = 0.3$ | CPS $\eta = 0.5$ | CPS $\eta = 0.7$ |
|---|---|---|---|---|---|
| Reward | 0.966 | 0.970 | 0.973 | 0.963 | 0.975 |

Reformulate it into the factorized form,

$$\boldsymbol{x}_t = \left( \frac{\sigma_t}{\sigma_s} \alpha_s e^{-h} + \alpha_t (1 - e^{-2h}) \right) \hat{\boldsymbol{x}}_0 + \sigma_t e^{-h} \hat{\boldsymbol{x}}_1 + \sigma_t \sqrt{1 - e^{-2h}} \boldsymbol{\epsilon}$$

$$= \left( \alpha_t e^{-h} e^{-h} + \alpha_t (1 - e^{-2h}) \right) \hat{\boldsymbol{x}}_0 + \sigma_t e^{-h} \hat{\boldsymbol{x}}_1 + \sigma_t \sqrt{1 - e^{-2h}} \boldsymbol{\epsilon}$$

$$= \alpha_t \hat{\boldsymbol{x}}_0 + \sigma_t e^{-h} \hat{\boldsymbol{x}}_1 + \sigma_t \sqrt{1 - e^{-2h}} \boldsymbol{\epsilon}. \tag{24}$$

The coefficient of sample is $\alpha_t$, which exactly matches the first condition of CPS. The total noise level is,

$$\sigma_{total} = \sigma_t \sqrt{e^{-2h} + 1 - e^{-2h}} = \sigma_t. \tag{25}$$

Thus, the SDE-DPM-Solver++1 perfectly matches the requirements of CPS. As also verified in (Lu et al., 2025), it is a special case of DDIM with $\eta = \sigma_t \sqrt{1 - e^{-2h}}$. Our proposed Flow-CPS can be seen as a special case of DDIM with $\eta = \sigma_t \sin(\frac{\eta \pi}{2})$ , which retains the hyper-parameter $\eta$ to tune the injected noise level. [2]

For higher-order DPM-Solvers, the high-order terms are residuals of two successive estimations of noise or sample, such as $\sigma_t (e^h - 1) \frac{\epsilon_\theta(\boldsymbol{x}_r, r) - \epsilon_\theta(\boldsymbol{x}_s, s)}{r_1}$. Since the coefficients of the two estimations cancel each other out, our analysis above remains unaffected for the higher-order DPM-Solvers.

Figure 8 illustrates the training curves for both Flow-CPS and SDE-DPM-Solver++1 utilizing the OCR reward. We observe that Flow-CPS becomes unstable when $\eta \leq 0.5$, characterized by intermittent and sudden drops in reward. Similarly, the training curves for SDE-DPM-Solver++1 exhibit comparable sudden drops, which typically correlate with a lack of diversity. In the right panel of Figure 8, we plot the injected noise level of SDE-DPM-Solver++1 alongside the equivalent $\eta$ in Flow-CPS. Notably, the equivalent $\eta$ for SDE-DPM-Solver++1 varies across timesteps: it initiates at $1.0$, gradually decreases to approximately $0.43$, and subsequently returns to $1.0$. The underlying cause of the significant instability observed in SDE-DPM-Solver++1 remains under investigation and requires further research.

This training instability leads to inconsistency in the final performance. As shown in Table 5, we conducted two separate experiments using SDE-DPM-Solver++1. The peak rewards for these runs vary from $0.966$ to $0.970$, highlighting the variance in outcomes. Such instability is detrimental to reproducibility and necessitates multiple trials to obtain a satisfactory model.

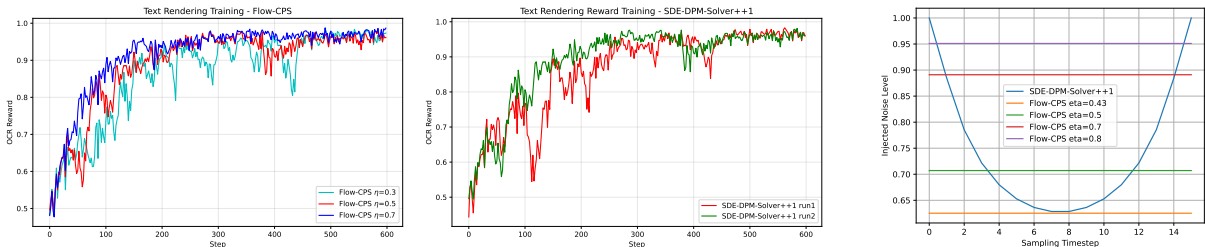

*Figure 8.* **Left**: The training curves of Flow-CPS with the OCR reward. **Middle**: The training curves of SDE-DPM-Solver++1 with the OCR reward. **Right**: The equivalent $\eta$ value for SDE-DPM-Solver++1.

## C. An Alternative for the numerical problem

In section 4.4, we mentioned that Flow-SDE has a numerical problem because of the $\frac{1}{t}$ term. Considering the limit of $\sigma_t \sqrt{\Delta t} \ll t - \Delta t$ and $\Delta t \to 0$ in Theorem 4.2, one possible patch would be setting $\sigma_t = \eta(t - \Delta t)$. Based on the Formula

---

[2] Here we swap the $\eta$ and $\sigma_t$ in the main text to follow the mathematical notations in DPM-Solver++.

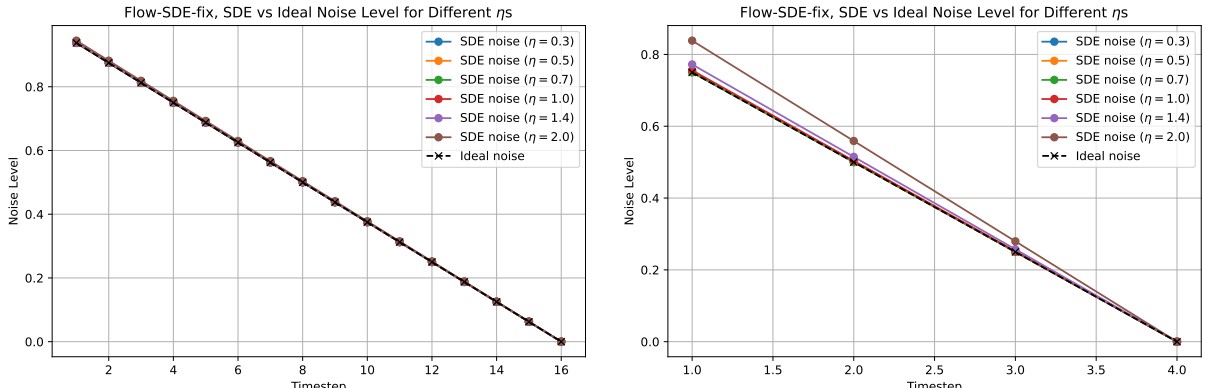

*Figure 9.* The ideal and SDE noise level for Equation 27. The error is ignorable when $\eta \leq 1$ for 4 and more steps.

18, our modified reverse Flow-SDE becomes,

$$\boldsymbol{x}_{t-\Delta t} \approx (1 - (t - \Delta t)) \hat{\boldsymbol{x}}_0 + \left(t - \Delta t - \frac{\eta^2}{2}(t - \Delta t)\Delta t\right)\hat{\boldsymbol{x}}_1 + \eta(t - \Delta t)\sqrt{\Delta t}\boldsymbol{\epsilon} \tag{26}$$

$$= \boldsymbol{x}_t - \hat{\boldsymbol{v}}_\theta(\boldsymbol{x}_t, t)\Delta t - \frac{\eta^2}{2}(t - \Delta t)\Delta t\hat{\boldsymbol{x}}_1 + \eta(t - \Delta t)\sqrt{\Delta t}\boldsymbol{\epsilon}$$

$$\approx \boldsymbol{x}_t - \hat{\boldsymbol{v}}_\theta(\boldsymbol{x}_t, t)\Delta t - \frac{\eta^2 t}{2}\hat{\boldsymbol{x}}_1 \Delta t + \eta t\sqrt{\Delta t}\boldsymbol{\epsilon}. \tag{27}$$

The noise level of Equation 26 is $(t - \Delta t)\sqrt{1 + (\frac{\eta^2 \Delta t}{2})^2}$, which is slightly higher than $(t - \Delta t)$. In the limit of $\Delta t \to 0$, Equation 27 converges to the following reverse SDE,

$$\mathrm{d}\boldsymbol{x}_t = \hat{\boldsymbol{v}}_\theta(\boldsymbol{x}_t, t)\mathrm{d}t + \frac{\eta^2 t}{2}\hat{\boldsymbol{x}}_1(\boldsymbol{x}_t, t)\mathrm{d}t + \eta t\mathrm{d}w. \tag{28}$$

Even though this formula still cannot meet the requirements of CPS, it has a smaller error than the original Flow-SDE. We show the noise level in Figure 9 and sampled images in Figure 10. It would be useful when the characteristics of the SDE are necessary.

| noise magnitude | 0 | 0.3 | 0.5 | 0.7 | 0.9 | 1.0 |
|---|---|---|---|---|---|---|

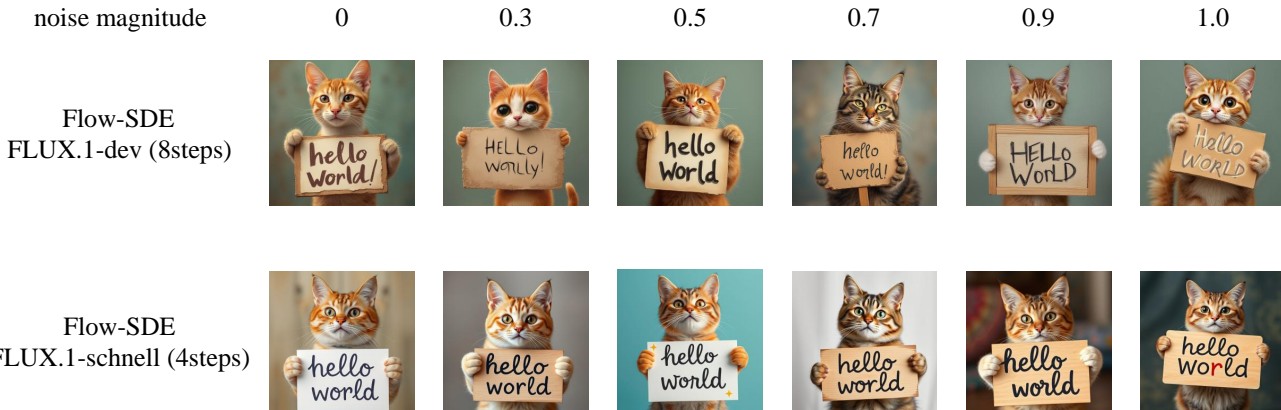

*Figure 10.* Image sampled by Equation 26 with $\eta = 1$. There is no obvious noise on these images.

## D. Qualitative Results

Figures 11 and 12 show visualizations of images optimized by the PickScore and HPSv2 reward models, respectively. Honestly speaking, a higher reward score does not necessarily equate to superior image quality. Often, the optimized images contain an excessive amount of detail, a phenomenon that can be seen as a way to "hack" the reward model. In practice, a balance must be found between achieving a high reward score and maintaining the image's visual coherence.

| Text prompt | FLUX baseline | Flow-GRPO PickScore | Flow-CPS PickScore |
|---|---|---|---|
| a 1980s japanese propaganda poster of the joker featured on artstation | 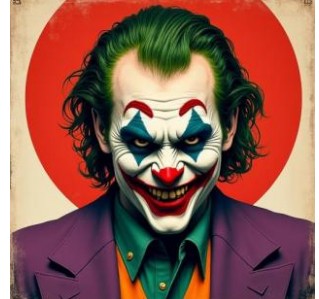 | 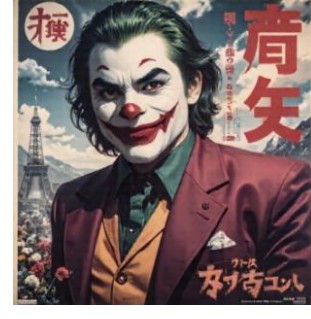 | 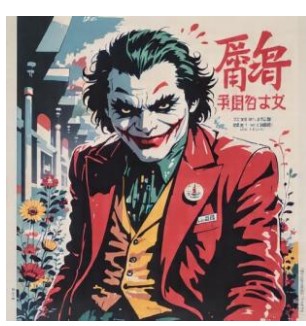 |
| digital art of a smiling frog in a tuxedo holding a glass of champagne | 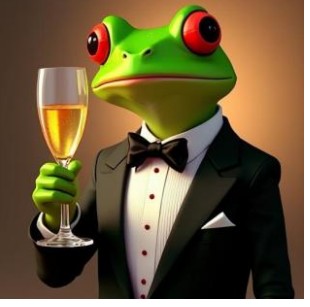 | 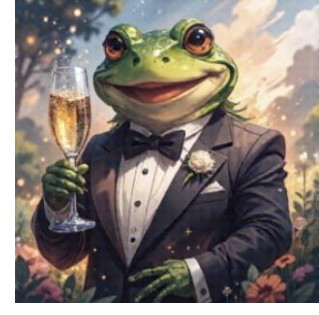 | 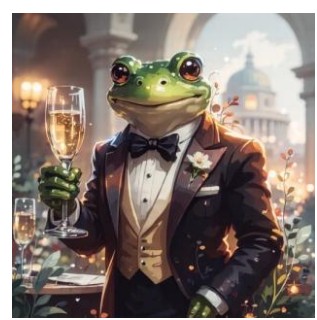 |
| instagram model working at an oilrig, covered in black oil, selfie, wearing hardhat | 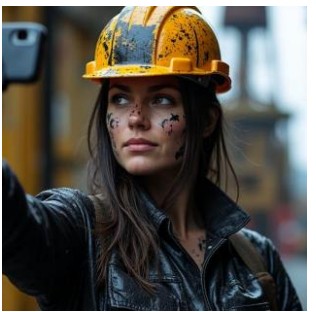 | 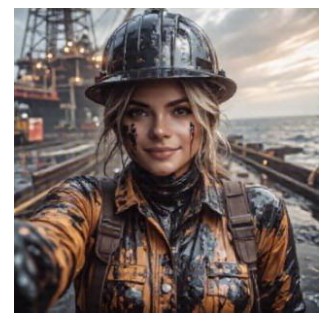 | 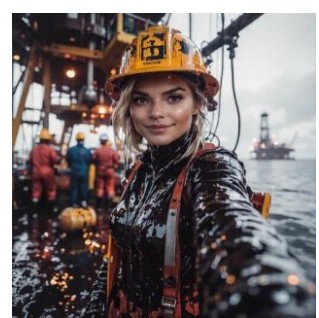 |
| oil painting of royal bearded dragon on gold throne with diamond crown | 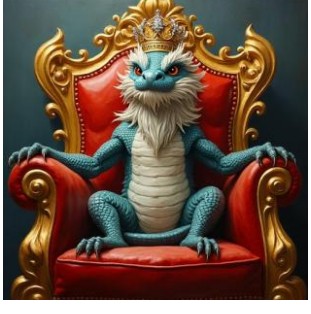 | 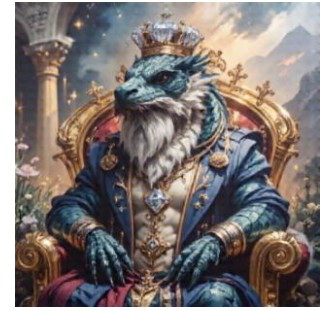 | 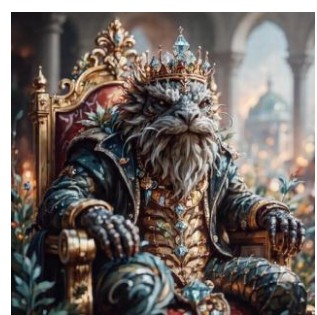 |
| a snowy chicago street during christmas art by ludwig fahrenkrog | 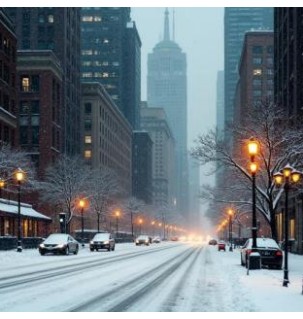 | 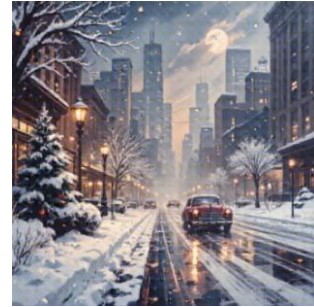 | 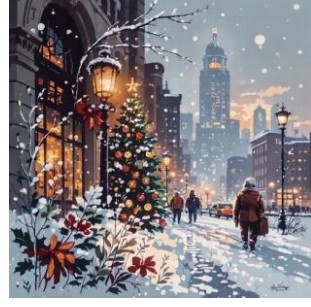 |
| a photo of furry teddy bears looking at a rover 75v8 car that is in the jungle, wideangle mgzt | 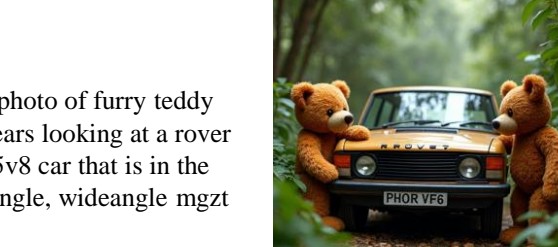 | 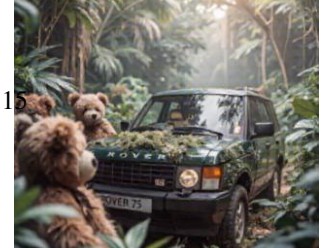 | 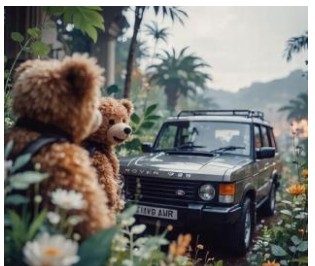 |

| Text prompt | FLUX baseline | Dance-GRPO HPSv2 | Flow-CPS HPSv2 |
|---|---|---|---|
| a bath room with a stand up shower and a bath tub. | 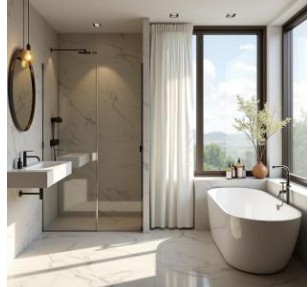 | 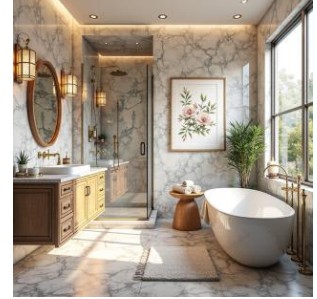 | 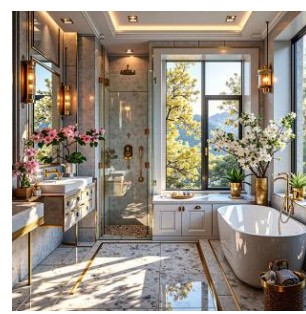 |
| A brown cow wearing yellow sunglasses in a pastel chalk drawing. | 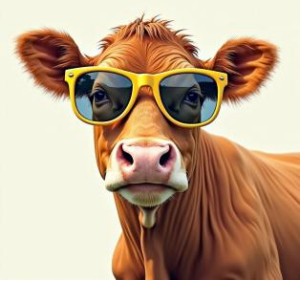 | 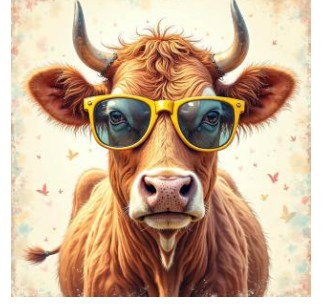 | 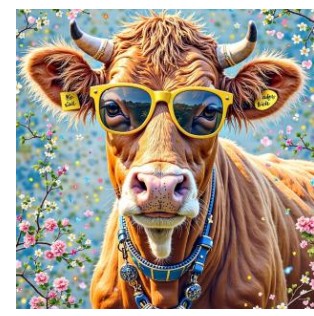 |
| Close-up portrait of a porcelain cyborg warrior with a biomechanical headdress and cyberpunk brackets. | 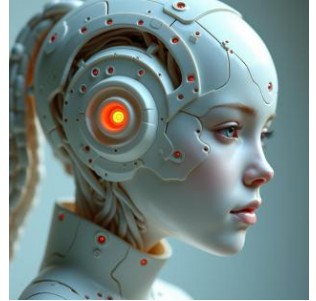 | 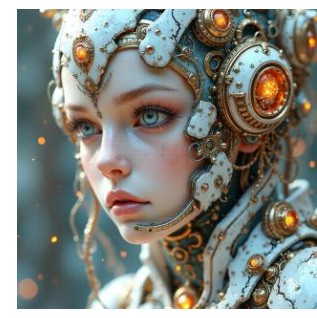 | 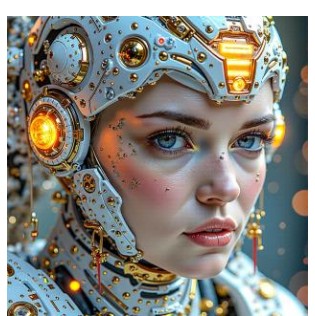 |
| Grunge painting of an empty road with a distant forest. | 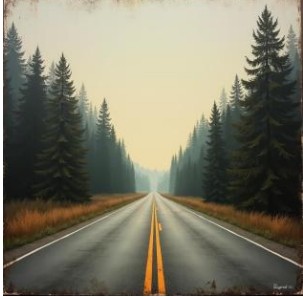 | 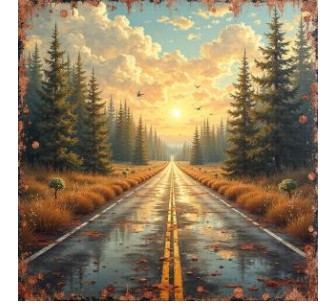 | 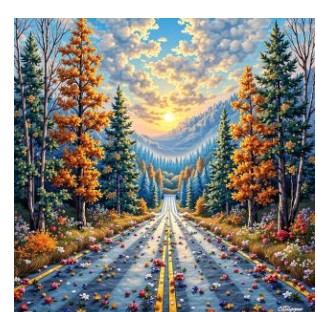 |
| The image is a digital art piece of Herr Gringleflapper's secret stash box with psychedelic and hypnotic effects. | 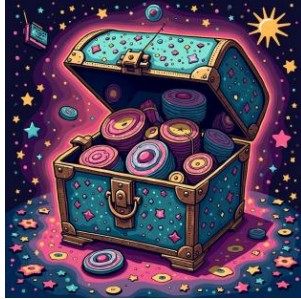 | 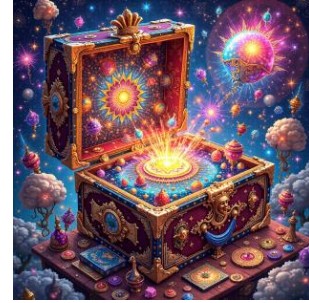 | 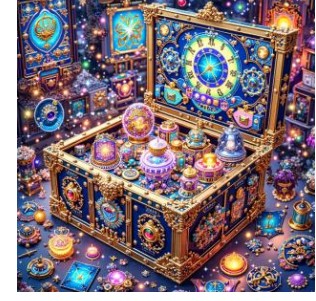 |
| An artwork from Dan Mumford collection featuring a mage invoking divine gods during a storm with | 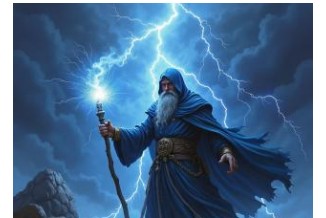 | 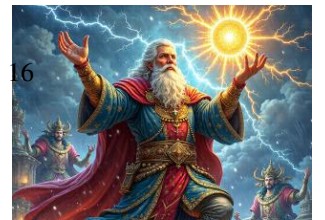 | 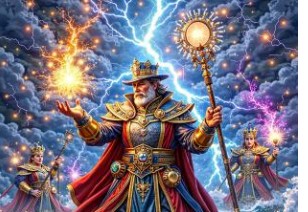 |