# OpenReview forum: "Coefficients-Preserving Sampling for Reinforcement Learning with Flow Matching"
_ICML.cc/2026/Conference — Submitted to ICML 2026_

### Official Review · Reviewer_Lj98 · 2026-03-03

**Soundness:** 3
**Presentation:** 3
**Significance:** 3
**Originality:** 2
**Overall Recommendation:** 4
**Confidence:** 4

**Summary:**

This paper studies the noise artifacts in SDE-based sampling for flow matching RL fine-tuning. The main challenge is that existing Flow-SDE formulations produce severely noisy images due to numerical instability near boundary conditions. To address this, the authors propose a coefficients-preserving sampling method that generalizes DDIM to flow matching, ensuring noise levels remain consistent with the scheduler at every timestep. The proposed method is evaluated on FLUX.1-dev and FLUX.1-schnell across multiple reward functions, demonstrating its effectiveness in eliminating noise artifacts and achieving faster RL convergence.

**Compliance With Llm Reviewing Policy:**

Affirmed.

**Key Questions For Authors:**

1. Do the noise artifacts from Flow-SDE also affect standard (non-RL) generation, or are they specific to the RL exploration setting?
2. How sensitive is the proposed method to the choice of noise level parameter $\eta$ across different reward functions?

**Limitations:**

yes

**Strengths And Weaknesses:**

**Strengths**

1. The proposed method is well motivated, since the noise artifacts from Flow-SDE are clearly visible (Figure 1) and the root cause is rigorously analyzed.
2. The paper is well written and easy to follow. The relationship between the existing Flow-SDE and the proposed formulation is clearly presented.

**Weaknesses**

1. The technical contribution seems incremental. DDIM-style sampling is well-established in diffusion [1]; the main novelty is adapting it to flow-based RL, and the fix follows naturally from the DDIM construction.
2. Lack of quantitative diversity evaluation. The authors argue that the proposed method maintains stochasticity for RL exploration but do not measure this (e.g., pairwise LPIPS, intra-group variance). Since diversity is critical for RL-based optimization, the authors should report diversity metrics to validate that exploration is preserved.
3. Incomplete experimental results. The paper evaluates on both FLUX and SD3.5 but does not provide full results across all model-dataset combinations. The authors should report complete results on each model for every dataset to allow fair comparison.

---

> ### Author Rebuttal · Authors · 2026-03-27
>
> We sincerely appreciate your insightful comments and efforts in reviewing our manuscript. We respond to each of your comments in what follows.
>
> > [w1] The technical contribution seems incremental. DDIM-style sampling is well-established in diffusion [1]; the main novelty is adapting it to flow-based RL, and the fix follows naturally from the DDIM construction.
>
> Yes, our proposed method is a noise-level-tunable variant of DDIM. Notably, our formulation avoids the numerical instability such as the issue of negative values under the square root and the small value under denominator.
>
> We emphasize that the core contribution of this work lies in the systematic discovery and theoretical diagnosis of the "over-noising" phenomenon in Flow-SDE. Through Theorem 1, we rigorously identify the root cause as the error accumulated from the first-order Taylor expansion during the SDE derivation.
>
> This paper follows a principled "problem discovery $\rightarrow$ theoretical analysis $\rightarrow$ solution" trajectory. Once the mathematical root of the "over-noising" problem is exposed, the resulting solution is naturally straightforward. If DDIM already solves this problem, why should we over-design a new algorithm for the sake of artificial novelty? We believe that such a theoretically grounded and transparent analysis is the most valuable part of our paper, and it aligns perfectly with the ICML ethos.
>
> > [w2] Lack of quantitative diversity evaluation.
>
> We omitted a diversity comparison because our baseline (Flow-SDE) employs insufficient denoising, leading to inherently high sample variety due to the excess noise. This stochasticity complicates the selection of a proper metric that can distinguish between meaningful diversity and sampling noise. Consequently, providing a rigorous and informative diversity assessment under these conditions proved to be non-trivial.
>
> We have done an experiment by fixing the prompt and initial seed, generating 100 samples, and measuring the diversity as the average Euclidean distance from the latents to their centroid. The results are as follows:
>
> | Method | 0 | 0.1 | 0.3 | 0.5 | 0.7 | 1.0 |
> | :--- | :---: | :---: | :---: | :---: | :---: | :---: |
> | **Flow-SDE** | 0 | 54.53 | 91.31 | 120.18 | 150.92 | 210.14 |
> | **Flow-CPS** | 0 | 61.63 | 103.99 | 128.36 | 142.57 | 150.88 |
>
> Crucially, we argue that the seemingly high diversity of Flow-SDE at $\eta=1.0$ is largely driven by spurious diversity (i.e., added Gaussian noise/artifacts) rather than meaningful semantic or textural variations. We posit that for RL to be effective, diversity must be grounded in image semantics and textures, not in high-frequency noise.
>
> In the regime of $\eta<0.5$, where images remain relatively clean, Flow-CPS exhibits slightly higher diversity. This marginal increase is primarily attributed to the removal of the $\sqrt{\Delta t}$ term before the injected noise. Ultimately, the core advantage of Flow-CPS is not merely inflating raw diversity metrics, but generating cleaner images that yield more accurate rewards, thereby facilitating better optimization.
>
> > [w3] Incomplete experimental results.
>
> We choose these combinations for the purpose of comparative fairness, as they are sourced directly from the baselines' original codebase. Diverging from these established configurartions could compromise the integrity of the head-to-head comparison. While the short duration of the rebuttal phase precludes us from completing all requested experiments now, we will report additional results across a broader range of settings in the final manuscript.
>
> > [q1] Do the noise artifacts from Flow-SDE also affect standard (non-RL) generation, or are they specific to the RL exploration setting?
>
> We clarify that the noise artifacts of Flow-SDE are not exclusive to RL settings; they are an intrinsic flaw that manifests in general generation tasks as well. As far as we know, Flow-SDE has not seen widespread adoption, and its absence from major libraries such as diffusers likely explains why this critical issue has remained under-explored. By identifying this universal artifact, our work fills a significant gap in the understanding of Flow-based SDE samplers, providing the first rigorous analysis of a problem that affects the broader utility of this sampling class.
>
> > [q2] How sensitive is the proposed method to the choice of noise level parameter $\eta$ across different reward functions?
>
> Our experiments reveal that the optimal $\eta$ is sensitive to the choice of reward: $\eta=0.9$ yields the best results for PickScore, while $\eta=0.7$ is preferable for OCR and GenEval. Currently, we treat $\eta$ as a tunable hyperparameter because its optimal setting appears to depend on the specific characteristics of the target task. We believe that providing this tunability is a significant advantage, as it allows for finer alignment with different reward signals compared to rigid, non-parameterized samplers.

---

> > ### Author Rebuttal · Reviewer_Lj98 · 2026-04-01
> >
> > The authors addressed my concerns with additional experimental results.

---

> > > ### Author Response · Authors · 2026-04-02
> > >
> > > Thank you for the positive confirmation! We are glad that the additional experimental results addressed your concerns. Since you have no further questions, we would be very grateful if you might consider raising your score.

---

### Official Review · Reviewer_af8J · 2026-03-12

**Soundness:** 3
**Presentation:** 2
**Significance:** 2
**Originality:** 2
**Overall Recommendation:** 3
**Confidence:** 4

**Summary:**

The paper studies RL-based methods for fine-tuning flow and diffusion models. It hypothesizes that the SDE-based sampling method that Flow-GRPO methods uses is suboptimal and proposes instead to DDIM-like methods. It motivates the use of DDIM-based sampling by it being “coefficient-preserving”. Experiments showcase the method on various tasks and setups.

**Compliance With Llm Reviewing Policy:**

Affirmed.

**Final Justification:**

The fundamental concerns I have about this work were unfortunately not addressed. The definition of coefficient-preserving are still not given in a mathematically rigorous fashion. Further, using ancestral sampling/DDIM sampling for SDEs in the context of RL fine-tuning (see discussions below) is a reasonable, yet not significantly contribution, in my view. However, I acknowledge that this is a judgement call and others might have different views.

**Key Questions For Authors:**

- How is the proposed method different from DDIM sampling with Flow-GRPO? Why is the definition of coefficient-preserving required given that DDIM sampling fulfills it?

**Limitations:**

Yes.

**Strengths And Weaknesses:**

Strengths:
- Stochasticity and exploration is a fundamental requirement for successful RL with flow and diffusion models. Therefore, it is reasonable to study the optimal sampling procedure.
- Experiments show improvements by the proposed method.

Weaknesses:
- The coefficient-preserving property and proposed sampling procedure seems to be a special case of DDIM sampling. Therefore, what the method effectively proposes to do is to combine DDIM sampling with Flow-GRPO and related methods. This should be stated more explicitly. The paper leaves it open whether the sampling method in itself is novel. Compare equation (12) in the paper with equation (12) in the DDIM paper (https://arxiv.org/pdf/2010.02502). By plugging in the terms, it seems like one recovers the other (note: time conventions are different across papers)? In particular, all DDIM sampling procedures for any alpha_t, sigma_t’s are coefficient-preserving based on the definition (see equation (12) in DDIM paper). Therefore, it is unclear why the definition of coefficient-preserving is needed.
- The definition of coefficient-preserving is very vague and should be stated more formally. For example, “The coefficient of the sample should be strictly allocated by the scheduler for all timesteps” - what does "allocated refer to here?
- Some parts of the paper should be stated more rigorously. For example, equation (3) should use a Wiener process (and not an epsilon) to describe the SDE.
- In equation (17), there is no Wiener process although it is stated.

---

> ### Author Rebuttal · Authors · 2026-03-27
>
> We sincerely appreciate your insightful comments and efforts in reviewing our manuscript. We respond to each of your comments in what follows.
>
> > [w1&q1] The coefficient-preserving property and proposed sampling procedure seems to be a special case of DDIM sampling. How is the proposed method different from DDIM sampling with Flow-GRPO? Why is the definition of coefficient-preserving required given that DDIM sampling fulfills it?
>
> We clarify that our method is a noise-level-tunable variant of DDIM. Additionally, our formulation avoids the numerical instability caused by negative values under the square root and very small value in the denominator. We have consistently highlighted this connection throughout the Abstract, Introduction, and Solution sections, explicitly saying that ”Inspired from DDIM”, ”generalizing DDIM to Flow Matching”, “referring to DDIM”. Our formulation derivation is exactly started from DDIM in Eq. (7).
>
> However, if this work were only about "combine DDIM with Flow-GRPO", we would not even start writing this paper. We wish to claim that our primary contribution is the discovery and theoretical diagnosis of the "over-noising" issue in Flow-SDE, which is used in two previous Diffusion-RL works. We identify the root cause as the error introduced by the first-order Taylor expansion during SDE derivation (Theorem 1).
>
> This paper follows a "problem identification $\rightarrow$ theoretical analysis $\rightarrow$ solution" trajectory. Once the theoretical root is exposed, the solution—a refined DDIM variant—is naturally elegant and simple. We do not want to add any artifical novelty on it to destroy the simplicity. We believe this type of theoretical analysis aligns perfectly with the ICML ethos.
>
> Furthermore, by formalizing the CPS conditions, which were implicit but never explicitly defined in the original DDIM paper, we provide a clear theoretical principle to help future researchers avoid similar noise-injection pitfalls as Flow-SDE.
>
> > [w2] The definition of coefficient-preserving is very vague and should be stated more formally. For example, “The coefficient of the sample should be strictly allocated by the scheduler for all timesteps” - what does "allocated refer to here?
>
> At each timestep $t$, the scheduler computes specific values for $\alpha_t$ and $\sigma_t$. The latents are then denoised to ensure that the sample coefficient aligns with $\alpha_t$ and the noise coefficient with $\sigma_t$. Consequently, the scheduler can be viewed as performing coefficient **allocation**, defining the precise ratio of signal to noise that the sampler must maintain throughout the trajectory.
>
> >[w3&w4] Some parts of the paper should be stated more rigorously. For example, equation (3) and equation (17) should use a Wiener process (and not an epsilon) to describe the SDE.
>
> Thanks for your advice. To explicitly denote the underlying Wiener process, we will replace the discrete representation $\sqrt{\Delta t} \epsilon$ with the continuous form $dW$. This notation ensures that our theoretical analysis is consistent with the continuous-time SDE framework, while $\sqrt{\Delta t} \epsilon$ remains the corresponding numerical implementation.

---

> > ### Author Rebuttal · Reviewer_af8J · 2026-03-31
> >
> > Thank you for your response. I will not increase my score further for the following reason: Even for the SDE that Flow-GRPO considers, one can do it in a coefficient-preserving fashion. This has nothing to do with the SDE but rather with the numerical solver used. This can be done via *ancestral sampling* (see [1] for example) which corresponds to a different numerical solver of the same ODE. Ancestral sampling is equivalent to DDPM sampling [2] (see explanations in [1]) with SDE/Flow models. Therefore, what this work effectively proposes is to use well-established sampling procedure (SDE solver) of ancestral sampling/DDIM sampling for GRPO of flow/diffusion models.  Unfortunately, I do not consider this significant contribution.
> >
> > Unfortunately, the definition of coefficient-preserving is still not given in a rigorous mathematical fashion. I strongly recommend given a *rigorous* definition. This will strengthen your work (this won't change my rating though).
> >
> > [1] Song, Yang, et al. "Score-based generative modeling through stochastic differential equations." arXiv preprint arXiv:2011.13456 (2020).

---

> > > ### Author Response · Authors · 2026-04-01
> > >
> > > Thank you for your professional response. We respectfully express our regret that you remain unconvinced by the contributions of our paper.
> > >
> > > 1. **The primary contribution of this paper lies in diagnosing the fundamental root cause of the over-noising issue:** the approximation errors introduced by the frequent use of Taylor expansions during the SDE derivation. Typically, once a problem's root cause is exposed, the resulting solution appears straightforward. In recent years, numerous well-performing samplers have been developed that do not suffer from this over-noising issue. While simply swapping in an existing sampler—whether DDIM, Ancestral Sampling (DDPM), or another stochastic variant—is trivial, doing so merely treats the symptom. It fails to answer the fundamental questions: Why did the Flow-SDE formulation make this over-noising error, and how can we guarantee we do not repeat this mistake in the future? We always believe that **"raising a problem is more essential than its solution"**.
> > >
> > > 2. **Revisiting the classic work by Song et al. (2020):** The continuous SDE is fundamentally an approximation of the discrete DDPM process, or vice versa (as shown in their Appendix B, using Taylor expansions). Furthermore, when deriving Ancestral Sampling in their Appendix E, the authors employed a similar three-step approximation from the reverse SDE solver. All approximations have errors and conditions under which those errors are negligible. Why were they able to navigate these approximations precisely targeting DDPM, but not opposite direction? Because they possessed the discrete DDPM formulation as a **"principled guidance"**—they already knew the exact mathematical destination. In the domain of Flow Matching with stochasticity, the absence of such a guidance is exactly what led to the approximation errors in Flow-SDE. The CPS definition proposed in our paper is designed to provide this missing **"principled guidance"**, which were implicit but never explicitly defined in the original DDIM paper, ensuring that an approximated SDE solver can be rigorously restored to an error-free discrete form. As described in our Eq. (18), through a two-step approximation, Flow-SDE can be mathematically transformed back into the Flow-CPWS.
> > >
> > > 3. **Ancestral Sampling (DDPM), much like the SDE-DPM-Solver++1 discussed in our appendix, relies on a pre-defined, fixed noise injection schedule.** In the context of Reinforcement Learning (RL) training, the inability to tune the stochasticity strength is a significant drawback, as it limits the ability to explicitly balance exploration and exploitation.
> > >
> > > 4. **Finally, we appreciate your critique regarding the mathematical rigor of our CPS definition.** We agree that our previous description was implementation-centric. In the revised manuscript, we will provide a strict formal definition grounded in marginal distributions:
> > >
> > > **[Rigorous CPS Definition]** A discrete sampling step $\hat{x}_{t-\Delta t} = f(\hat{x}_t, t, \Delta t, \epsilon)$ is Coefficient-Preserving if and only if it exactly satisfies the target marginals for any timestep $t$ and any valid step size $\Delta t$:
> > >    * (1) $\mathbb{E}[\hat{x}_{t-\Delta t} | \hat{x}_0] = \alpha\_{t-\Delta t} \hat{x}\_0$
> > >    * (2) $\text{Var}(\hat{x}_{t-\Delta t} | \hat{x}_0) = \sigma\_{t-\Delta t}^2 I$
> > >
> > > where $\alpha_{t-\Delta t}$ and $\sigma_{t-\Delta t}$ are defined by the specific algorithm. For Flow Matching, $\alpha_t = 1 - t$ and $\sigma_t = t$.
> > >
> > > Once again, we genuinely appreciate the time and effort you have dedicated to reviewing our work. While we respect your decision regarding the score, your rigorous critique has undeniably helped us refine our mathematical definitions and strengthen the theoretical foundation of our paper. We believe the formalized CPS framework now presents a solid and necessary principle for the community.

---

### Official Review · Reviewer_gdw5 · 2026-03-13

**Soundness:** 3
**Presentation:** 3
**Significance:** 3
**Originality:** 3
**Overall Recommendation:** 4
**Confidence:** 4

**Summary:**

This paper observes that previous ODE-to-SDE methods suffer from noisy sampling. Through a coefficients-preserving analysis, the authors point out that the core issue is a mismatch between the denoising and noising scales. Based on this analysis, the authors propose a new sampling method to improve sampling quality, and validate its effectiveness in Reinforcement Learning (RL).

**Compliance With Llm Reviewing Policy:**

Affirmed.

**Final Justification:**

Thank you for the authors’ response. I have no further questions.

**Key Questions For Authors:**

I hope the authors can further improve the discussions in the paper, as I have listed in the weaknesses section.

**Limitations:**

I did not see a clear discussion of limitations from the authors. However, I actually think this work does not have any glaring limitations. If I had to point one out, it would be the lack of sufficient comparison with dance-grpo, making it unable to demonstrate its advantages over it. There might be some performance shortcomings in practical applications.

**Strengths And Weaknesses:**

1. Strengths:
    * The problem studied in this paper is highly important in RL. The authors' analysis is simple and clear, successfully pointing out the core issue of previous methods.
    * The authors validate the effectiveness of GRPO + CPS in RL.
    * The writing is clear and easy to follow.
2. Weaknesses:
    * It seems that CPS might affect diversity, but the authors do not appear to discuss this.
    * As a sampling method itself, CPS lacks experimental evaluation specifically focused on sampling. The authors should set up experiments to directly compare the sampling quality between previous methods and CPS.
    * From Figure 2, the issues with dance-grpo seem much less severe than those with flow-grpo. Except for the right side of Figure 5, which compares dance-grpo, all other experiments only compare flow-grpo. Is this because, in terms of performance, CPS does not have a clear advantage over dance-grpo?

---

> ### Author Rebuttal · Authors · 2026-03-27
>
> We sincerely appreciate your insightful comments and efforts in reviewing our manuscript. We respond to each of your comments in what follows. The following analysis and experiments will be updated into the paper in the next revision.
>
> > [w1] It seems that CPS might affect diversity, but the authors do not appear to discuss this.
>
> We omitted a diversity comparison because our baseline (Flow-SDE) employs insufficient denoising, leading to inherently high sample variety due to the excess noise. This stochasticity complicates the selection of a proper metric that can distinguish between meaningful diversity and sampling noise. Consequently, providing a rigorous and informative diversity assessment under these conditions proved to be non-trivial.
>
> We have done an experiment by fixing the prompt and initial seed, generating 100 samples, and measuring the diversity as the average Euclidean distance from the latents to their centroid. The results are as follows:
>
> | Method | 0 | 0.1 | 0.3 | 0.5 | 0.7 | 1.0 |
> | :--- | :---: | :---: | :---: | :---: | :---: | :---: |
> | **Flow-SDE** | 0 | 54.53 | 91.31 | 120.18 | 150.92 | 210.14 |
> | **Flow-CPS** | 0 | 61.63 | 103.99 | 128.36 | 142.57 | 150.88 |
>
> Crucially, we argue that the seemingly high diversity of Flow-SDE at $\eta=1.0$ is largely driven by spurious diversity (i.e., added Gaussian noise/artifacts) rather than meaningful semantic or textural variations. We posit that for RL to be effective, diversity must be grounded in image semantics and textures, not in high-frequency noise.
>
> In the regime of $\eta<0.5$, where images remain relatively clean, Flow-CPS exhibits slightly higher diversity. This marginal increase is primarily attributed to the removal of the $\sqrt{\Delta t}$ term before the injected noise. Ultimately, the core advantage of Flow-CPS is not merely inflating raw diversity metrics, but generating cleaner images that yield more accurate rewards, thereby facilitating better optimization.
>
> > [w2] As a sampling method itself, CPS lacks experimental evaluation specifically focused on sampling. The authors should set up experiments to directly compare the sampling quality between previous methods and CPS.
>
> Regarding qualitative image quality, we believe the visual evidence in Figure 1 is compelling: while images sampled by Flow-SDE exhibit significant noise artifacts, our method produces results that are free of perceptible noise. Quantitatively, PickScore and HPSv2 serve as perceptual metrics grounded in human preference. Since human annotators consistently penalize visual artifacts such as noise, these metrics inherently favor clean, high-quality images. As demonstrated in Figure 4, Flow-CPS consistently achieves higher PickScore rewards than Flow-SDE, further validating its superior alignment with human visual preferences.
>
> To further quantify image quality, we employed the MUSIQ [1] model. We evaluated 100 randomly generated images across different sampling algorithms and noise levels (). The average results are reported below:
>
> | Method | $\eta=0$ | $\eta=0.3$ | $\eta=0.5$ | $\eta=0.7$ | $\eta=0.9$ | $\eta=1.0$ |
> | :--- | :---: | :---: | :---: | :---: | :---: | :---: |
> | **Flow-SDE** | 77.29 | 77.49 | 76.93 | 74.48 | 71.49 | 64.13 |
> | **Flow-CPS** | 77.30 | 77.45 | 76.94 | 76.90 | 76.28 | 76.51 |
>
> As demonstrated in the table, Flow-CPS maintains robust image quality across all noise levels. In sharp contrast, Flow-SDE suffers from significant degradation as stochasticity increases (dropping from 77.29 to 64.13 at ). This quantitative evidence confirms that our method effectively eliminates the noise-induced quality deterioration observed in the baseline.
>
> [1] Ke, Junjie, et al. "MUSIQ: Multi-scale Image Quality Transformer." ICCV 2021.
>
> > [w3] From Figure 2, the issues with dance-grpo seem much less severe than those with flow-grpo.
>
> This experimental setup was primarily chosen because the DanceGRPO codebase only provides configurations for HPS [2]. To ensure strict comparative fairness, we conducted all comparative experiments within the baseline’s original infrastructure, modifying only the sampler component. By keeping the rest of the pipeline identical to the baseline, we ensure that the observed performance gains are directly attributable to our method rather than discrepancies in implementation or environment.
>
> We used to communicate with the authors of Dance-GRPO, and confirmed that they also noticed the over-noising issue. Their mitigation strategy involves a heuristic approach: reducing the noise level to 0.3 and omitting the additional noise in the final step. While this succeeds in producing images without visible noise, it harms exploration, which ultimately leads to suboptimal RL performance. In contrast, our method addresses the over-noising problem fundamentally without sacrificing the exploration.
>
> [2] https://github.com/XueZeyue/DanceGRPO/blob/main/fastvideo/config_sd/dgx.py

---

> > ### Author Rebuttal · Reviewer_gdw5 · 2026-04-03
> >
> > I have no further questions.

---

> > > ### Author Response · Authors · 2026-04-03
> > >
> > > Thank you for the confirmation! Since you have no further questions, we would be very grateful if you might consider raising your score. Thank you again for your time and support!

---

### Official Review · Reviewer_EvYn · 2026-03-13

**Soundness:** 3
**Presentation:** 2
**Significance:** 3
**Originality:** 3
**Overall Recommendation:** 4
**Confidence:** 3

**Summary:**

The article explores how to optimize the sampling process in reinforcement learning generative models based on flow matching. The author found that the sampling method based on SDE produces serious noise artifacts, resulting in the generated image being filled with noise, and pointed out that the root of the problem lies in Flow-SDE injecting excessive noise beyond what the original ODE scheduler allows. To address this issue, the author proposed using Flow-CPS and improved the calculation of logarithmic probability. Finally, the experiment proved that Flow-CPS eliminated noise artifacts and achieved better results.

**Compliance With Llm Reviewing Policy:**

Affirmed.

**Final Justification:**

Thank the authors for the reply of my questions, I have no further question and will maintain my score.

**Key Questions For Authors:**

1. As SDE-DPM Solver++1 is an existing method that satisfies CPS, why not incorporate it into the experiment in main paper?
2. The author mentioned the impact of Reward Hacking in the appendix. Since Flow CPS is the ability to improve reward model scores faster and higher, it is facing higher risk of reward hacking than the baseline method. Can the author discuss this in detail?
3. Refer to weaknesses section.

**Limitations:**

Yes.

**Strengths And Weaknesses:**

Strength

1. The method is concise and elegant, and can directly replace the SDE module in the existing baseline, with low project implementation cost.
2. The CPS method in this article can still maintain perfect noise free performance under few step sampling, which is very in line with the current development direction of mainstream models.

Weaknesses

1. In most experiments, the author removed KL divergence in order to pursue training speed, but in the GenEval task, it had to be added because the model would crash. This raises doubts about the robustness of the algorithm.
2. Removing \sigma_t in (15) changes the unbiasedness of the policy gradient objective. Lack of dicussion about why it still ensures the correct direction of policy optimization.

---

> ### Author Rebuttal · Authors · 2026-03-27
>
> We sincerely appreciate your insightful comments and efforts in reviewing our manuscript. We respond to each of your comments in what follows.
>
> > [w1] In most experiments, the author removed KL divergence in order to pursue training speed, but in the GenEval task, it had to be added because the model would crash. This raises doubts about the robustness of the algorithm.
>
> In the majority of our experiments, we chose to disable the KL loss to ensure a **fair comparison**. This decision was motivated by the fact that the KL loss significantly impacts convergence speed, which is highly sensitive to the choice of the $\beta$ hyperparameter and the step size. By removing this term, we mitigate the confounding influence of hyperparameter tuning. Furthermore, we empirically observed that disabling the KL loss does not compromise the model’s final performance on most tasks, particularly those evaluating image quality. The inclusion of KL loss specifically for the GenEval task was necessary to prevent model collapse and achieve optimal results, as we found the training to be unstable without this regularizer in that specific context.
>
> > [w2] Removing $\sigma_t$ in (15) changes the unbiasedness of the policy gradient objective. Lack of dicussion about why it still ensures the correct direction of policy optimization.
>
> We were compelled to remove the $\sigma_t$ term to ensure numerical stability. As $\sigma_t$ resides in the denominator and approaches zero during the final denoising steps, it triggers severe numerical issues that lead to model collapse. Furthermore, we observed that omitting $\sigma_t$ significantly accelerates convergence. As analyzed immediately below Eq. (15), $\sigma_t$ essentially functions as a loss weighting factor that biases training toward the later stages of the diffusion process. However, the initial steps are more critical for capturing sample diversity. We provide empirical evidence for this design choice in Section 5.4 and Figure 7.
>
> > [q1] As SDE-DPM Solver++1 is an existing method that satisfies CPS, why not incorporate it into the experiment in main paper?
>
> While SDE-DPM Solver++1 also satisfies CPS, it relies on a predefined noise level scheduler. In contrast, our method allows for noise level tuning, which is vital for RL performance. We will move the SDE-DPM Solver++1 results to the main paper in the final version, while keeping the detailed technical analysis in the Appendix due to the page limit.
>
> > [q2] The author mentioned the impact of Reward Hacking in the appendix. Since Flow CPS is the ability to improve reward model scores faster and higher, it is facing higher risk of reward hacking than the baseline method. Can the author discuss this in detail?
>
> We acknowledge that while Flow-CPS achieves higher rewards, it does indeed lead to increased reward hacking. As illustrated in the last two figures of the Appendix, Flow-CPS generates more fine-grained details compared to the baseline; however, these details do not always align with human perceptual aesthetics, despite successfully maximizing the reward signal. In the current RL paradigm, reward hacking remains a pervasive and challenging problem, and providing a fundamental solution is beyond the scope of this paper. We will include a dedicated discussion regarding the reward hacking in Appendix D.

---

> > ### Author Rebuttal · Reviewer_EvYn · 2026-04-02
> >
> > Thank the authors for the detail response, my concerns and questions have been fully resolved, I have no further issues.

---

> > > ### Author Response · Authors · 2026-04-02
> > >
> > > Thank you for the positive confirmation. We are thrilled that our detailed response has fully resolved your questions. Since you have no further questions, we would be very grateful if you might consider raising your score.

---

### Decision · Program_Chairs · 2026-04-30

**Decision:**

Reject

**Comment:**

This submission diagnoses an over-noising issue in Flow-SDE sampling for RL fine-tuning of flow matching models and proposes Flow-CPS, a DDIM-style coefficients-preserving sampler. It received three Weak Accepts and one Weak Reject (af8J, C4).

Merits. The noise artifacts in Flow-SDE are clearly illustrated, and Flow-CPS yields consistent empirical gains across several reward functions on FLUX.1-dev and FLUX.1-schnell. Three reviewers marked their concerns as resolved.

Remaining concerns. After reviewing the full discussion, I find the technical concerns raised by af8J (C4) the most decisive and only partially resolved:

In the rebuttal the authors state that Flow-CPS is "a noise-level-tunable variant of DDIM." af8J further notes that coefficients-preserving stochastic sampling is already achievable via ancestral/DDPM sampling (Song et al., 2020). The methodological delta over these established samplers therefore appears limited.
The CPS property, which the paper positions as a conceptual contribution, is not formally defined in the submitted manuscript. A rigorous definition was only introduced in the final author response and is deferred to the camera-ready.
Experimental coverage is acknowledged to be incomplete: the DanceGRPO comparison is restricted to a single reward (HPS, flagged by gdw5); several model–dataset combinations are missing (flagged by Lj98); the algorithm required re-enabling KL regularization on GenEval to avoid collapse (flagged by EvYn).
Decision. The significance of the contribution — rather than its correctness — is the point of disagreement. As af8J notes, "this is a judgement call and others might have different views." Weighing the limited methodological novelty over existing stochastic samplers and the under-specified central definition in the submitted version against the clearly presented diagnostic and positive empirical results, I recommend Reject, and encourage resubmission once the CPS property is formalized and the comparisons against ancestral sampling / SDE-DPM-Solver++ and DanceGRPO are broadened.